# S-Equol as a Gut-Derived Phytoestrogen Targeting Estrogen Receptor β: A Promising Bioactive Nutrient for Bone Health in Aging Women and Men: A Narrative Review

**DOI:** 10.3390/nu17243962

**Published:** 2025-12-18

**Authors:** Akira Sekikawa, Ashley Weaver, Kelly Mroz, Nina Z. Heilmann, Diana A. Madrid Fuentes, Kristen J. Koltun, Lauren J. Carlson, Kristin L. Cattell, Mengyi Li, Jiatong Li, Timothy M. Hughes, Elsa Strotmeyer, Bradley Nindl, Jane A. Cauley

**Affiliations:** 1Department of Epidemiology, School of Public Health, University of Pittsburgh, Pittsburgh, PA 15213, USA; ljc25@pitt.edu (L.J.C.); mel228@pitt.edu (M.L.); jil292@pitt.edu (J.L.); strotmeyere@edc.pitt.edu (E.S.); jcauley@edc.pitt.edu (J.A.C.); 2Department of Biomedical Engineering, Wake Forest University School of Medicine, Winston-Salem, NC 27157, USA; asweaver@wakehealth.edu (A.W.); nina.heilmann@advocatehealth.org (N.Z.H.); diana.madridfuentes@wfusm.edu (D.A.M.F.); kristin.cattell@advocatehealth.org (K.L.C.); 3Neuromuscular Research Laboratory, School of Health and Rehabilitation Sciences, University of Pittsburgh, Pittsburgh, PA 15219, USA; kem324@pitt.edu (K.M.); kjk116@pitt.edu (K.J.K.); bnindl@pitt.edu (B.N.); 4Department of Internal Medicine, Section on Gerontology and Geriatric Medicine, Wake Forest University School of Medicine, Winston-Salem, NC 27157, USA; timothy.hughes@wfusm.edu

**Keywords:** S-equol, estrogen receptor β, soy isoflavones, gut microbiome, bone health, precision nutrition

## Abstract

**Background/Objectives**: Osteoporosis is highly prevalent and contributes substantially to morbidity and mortality, yet long-term concerns about pharmacologic therapies leave a major treatment gap. Soy isoflavones have been investigated as safer alternatives, but results across trials are inconsistent. A key unresolved issue is the equol-producer phenotype, the gut microbial ability to convert daidzein to S-equol, the most bioactive isoflavone metabolite, which may explain much of this variability. This narrative review synthesizes mechanistic, translational, and clinical evidence to clarify the potential skeletal relevance of S-equol. **Methods**: Literature was identified through PubMed and Scopus searches (January 2000–October 2025) for experimental, mechanistic, and clinical studies examining S-equol, estrogen receptor β (ERβ), and bone metabolism, with emphasis on equol-producing status, bone strength and bone microarchitecture. **Results**: S-equol acts as a high-affinity ERβ agonist with antioxidant and anti-inflammatory properties but lacks the carcinogenic or thrombotic risks linked to ERα activation. In estrogen-deficient rodent models, S-equol improves trabecular bone volume by 10–20%, increases trabecular number, and enhances biomechanical strength. These findings align with preclinical evidence demonstrating that S-equol preserves trabecular microarchitecture, enhances bone strength, and reduces bone turnover. A limited number of human trials show reductions in bone resorption by 20% at a daily dose of 10 mg S-equol. In contrast, trials of soy isoflavones in humans have produced inconsistent findings, partly because of substantial variability in equol-producer phenotype among participants and the reliance on dual-energy X-ray absorptiometry, which cannot distinguish trabecular from cortical compartments. Advanced bone imaging and microbiome-informed approaches enable the precise evaluation of S-equol’s skeletal effects on trabecular bone and cortical bone, separately. **Conclusions**: S-equol represents a promising model for “precision nutrition,” where microbiome, hormonal, and host factors converge with potential to prevent age-related bone fragility. Rigorous trials that integrate microbiome phenotyping and advanced imaging are needed to validate this approach, translate mechanistic promise into clinical benefit, and better define safety.

## 1. Introduction

Osteoporosis is a major global health problem characterized by progressive loss of bone mineral density (BMD) that begins as early as the third decade of life [1]. The lifetime risk of osteoporotic fracture after age 50 is 40–50% in women and 13–33% in men [2,3]. More than 2 million fractures occur annually, with associated healthcare costs now exceeding 25 billion US dollars [4]. Fractures often recur and are associated with high mortality and long-term disability [5,6,7], underscoring the urgent need for safe, non-pharmacologic strategies to prevent bone loss and fragility fractures in aging populations.

Bioactive dietary compounds are increasingly investigated for their capacity to modulate skeletal remodeling. In addition to supplying essential nutrients, many food proteins and peptides exert bioactive effects beyond basic nutrition, particularly when digestion, hydrolysis, or fermentation generates peptides that can act systemically [8]. Soy isoflavones, naturally occurring phytoestrogens structurally similar to 17β-estradiol, exert estrogen-receptor–mediated actions, but their clinical effects on bone have been inconclusive [9]. A key source of variability is the gut microbiome–dependent conversion of the parent isoflavone daidzein into S-equol, a metabolite with markedly greater biological potency [10]. The “equol-producer” phenotype varies across populations and over time within individuals, with higher prevalence generally observed in Asian cohorts than in Western cohorts [11]. Despite research across mechanistic, microbiome, and clinical domains, these bodies of evidence have rarely been integrated to explain why responses to isoflavones and S-equol differ across individuals and populations. This synthesis is needed to clarify how biological selectivity, phenotypic variability, and measurement limitations jointly shape observed outcomes.

Mechanistically, S-equol acts as a selective estrogen receptor beta (ERβ) agonist with minimal affinity for estrogen receptor alpha (ERα) [10]. ERβ is highly expressed in trabecular bone and mediates anti-resorptive, antioxidant, and anti-inflammatory effects, in contrast to ERα, which predominates in cortical bone and is responsible for classical estrogenic actions [12,13]. These ERβ-specific pathways may help explain why S-equol shows strong skeletal effects in preclinical studies yet inconsistent results in human trials, especially when equol-producer status is not assessed. Moreover, ERβ signaling exerts favorable effects on vascular endothelium and reduces oxidative stress, supporting an integrated model of skeletal and vascular protection. Importantly, ERβ activation lacks the carcinogenic and thrombotic risks associated with ERα stimulation [14], as demonstrated by the Women’s Health Initiative (WHI), in which systemic estrogen therapy reduced fracture risk but increased the incidence of stroke and venous thromboembolism [15]. Thus, S-equol and other ERβ-selective agonists offer a promising strategy for maintaining skeletal integrity while avoiding the potential adverse outcomes associated with postmenopausal hormone therapy.

Preclinical studies consistently show that S-equol prevents trabecular bone loss and enhances biomechanical strength in estrogen-deficient models through ERβ-mediated anti-resorptive, antioxidant, and cytoprotective pathways [16,17,18]. Human trials, although limited in size and duration, demonstrate a reduction in bone resorption markers and suggest more pronounced effects among equol producers [19,20,21,22]. These findings highlight the potential of S-equol as a strategy for supporting skeletal health in the aging population. These divergent findings underscore the importance of synthesizing evidence across mechanistic, microbiome, and imaging domains to clarify when, why, and for whom S-equol is likely to confer skeletal benefit.

Because dual-energy X-ray absorptiometry (DXA) cannot distinguish cortical from trabecular compartments or detect microstructural deterioration, its limitations have likely obscured compartment-specific skeletal responses to S-equol in prior trials. Emerging imaging modalities, such as high-resolution peripheral quantitative computed tomography (HR-pQCT) [23], offer a more refined assessment of bone quality, and future studies integrating these approaches with microbiome-informed phenotyping will be important for clarifying the skeletal relevance of S-equol.

These gaps highlight the need for an integrative synthesis that brings together ERβ biology, microbiome-dependent equol production, and emerging approaches to skeletal phenotyping. This review advances the field by proposing a unified framework to interpret heterogeneous findings across study designs and by identifying the mechanistic and methodological factors most likely to influence skeletal outcomes.

This review synthesizes mechanistic, translational, and clinical evidence on S-equol, an ERβ-selective, gut-derived nutrient, and its impact on skeletal health. We emphasize what DXA missed and how HR-pQCT and micro–finite element strength estimates can clarify compartment-specific effects. Finally, we identify key gaps (few long-term trials, underrepresentation of men, inconsistent phenotyping) and discuss priorities for next-generation studies to evaluate S-equol as a scalable precision nutrition strategy for bone fragility. The recent availability of S-equol as a dietary supplement [24] enables direct testing in clinical trials, independent of equol-producer phenotype.

## 2. Materials and Methods

Given the heterogeneity of study design and research questions, we conducted a narrative review rather than a systematic review. Targeted searches were performed in PubMed and Scopus (January 2000–October 2025) using combinations of the following terms: S-equol, soy isoflavones, ERβ, osteoporosis, DXA, BMD, bone microarchitecture, HR-pQCT, gut microbiome, and precision nutrition. We included experimental, translational, and clinical studies, as well as reviews and meta-analyses on isoflavones/S-equol and bone. Selection emphasized ERβ-related mechanisms, trabecular versus cortical effects, and the equol-producing phenotype. We included randomized controlled trials (RCTs), prospective and cross-sectional observational studies, animal models of estrogen deficiency, and in vitro mechanistic studies that examined S-equol, soy isoflavones, or related agents affecting ERβ signaling or microbial equol production. Eligible human studies enrolled postmenopausal women or men, while eligible preclinical studies used ovariectomized or otherwise estrogen-deficient rodent models. To be included, studies needed to report outcomes related to BMD, bone turnover markers, bone microarchitecture (e.g., HR-pQCT), biomechanical strength, fracture outcomes, or mechanistic endpoints pertinent to osteoclast and osteoblast biology or the daidzein-to-S-equol pathway. We excluded non–non-peer-reviewed sources, isolated case reports, and studies lacking bone- or mechanism-relevant outcomes. Evidence from animal models, RCTs, and observational studies was integrated narratively. Quality considerations prioritized studies with clearly defined equol-producer status and objective bone measures (e.g., BMD, HR-pQCT, bone turnover markers).

A narrative synthesis was selected for three reasons. First, the research question spans mechanistic, translational, and human clinical evidence, requiring an integrated interpretation that cannot be captured by quantitative pooling alone. Second, substantial methodological heterogeneity across the literature, including differences in isoflavone preparations, S-equol doses, intervention durations, imaging modalities, phenotyping approaches, and population characteristics, precludes a valid or meaningful meta-analysis. Third, our goal was to contextualize emerging mechanistic and microbiome insights with clinical findings to identify conceptual gaps, particularly regarding equol-producer phenotyping and advanced imaging, which is most appropriately addressed through a narrative framework rather than a systematic review.

## 3. Results

### 3.1. Mechanisms of Estrogen and ERβ in Bone

Estrogen maintains skeletal homeostasis by coordinating osteoclast resorption with osteoblast/osteocyte formation and maintenance. With aging and estrogen deficiency, this coupling is disrupted, leading to excessive remodeling, trabecular perforation, and cortical thinning [25,26]. These effects are mediated by two nuclear receptors (i.e., ERα and ERβ) expressed in bone cells but with distinct distribution and signaling profiles (Table 1) [25,26,27].

#### 3.1.1. Compartmental and Cellular Distribution

ERα and ERβ exhibit distinct spatial and cellular patterns within the skeleton that help explain their divergent functional roles. ERα signaling predominates in cortical envelopes, particularly along the periosteal and endocortical surfaces [12], where it regulates cortical thickness, periosteal expansion, and endocortical resorption. Declines in ERα activity after menopause [28] contribute to cortical porosity, thinning, and loss of bending strength, which are major determinants of fracture risk. In contrast, ERβ is broadly expressed across osteoblast-lineage cells, osteocytes, and marrow stromal cells, with relatively higher representation in trabecular-rich regions where metabolic turnover is greatest [12]. These compartmental differences suggest that ERβ-selective ligands such as S-equol may exert their greatest effects within trabecular bone, where ERβ-driven anti-resorptive and cytoprotective actions are most relevant and where microarchitectural deterioration progresses rapidly after menopause. This distributional framework provides the biological rationale summarized in Table 1, linking receptor localization to compartment-specific skeletal effects [25,26,27].

#### 3.1.2. Osteoclast Restraint via Genomic Signaling

Ligand-activated ERβ forms dimers that bind estrogen response elements or tether to other transcription factors to modulate genes governing osteoclastogenesis. Activation of ERβ upregulates osteoprotegerin (OPG) and suppresses Receptor Activator of Nuclear Factor-κB Ligand (RANKL) expression, reducing RANKL–RANK signaling and thereby limiting osteoclast differentiation and survival [12,27]. ERβ also dampens pro-resorptive cytokine pathways (e.g., Nuclear Factor kappa-light-chain-enhancer of activated B cells–dependent Tumor Necrosis Factor α/Interleukin-6 signaling), shifting the local milieu toward lower bone turnover [12,27].

#### 3.1.3. Redox and Inflammatory Control Supporting Formation

Estrogen deficiency heightens oxidative stress and inflammatory tone in bone, impairing osteoblastogenesis and promoting osteoclast activity. In vitro, S-equol suppresses osteoclastogenesis, promotes osteoblast differentiation, and modulates oxidative stress by activating Phosphoinositide 3-kinase/Protein kinase B (PI3K/Akt) and Nuclear factor erythroid 2–related factor 2 (Nrf2) signaling cascades [17,18]. ERβ activation engages pro-survival and antioxidant pathways, including cross-talk with PI3K/Akt, and dampens inflammatory signaling, helping preserve osteoblast function and viability and reducing osteoclast drive [12,27]. These actions are particularly relevant in trabecular bone, where high surface area renders microarchitecture sensitive to redox and cytokine perturbations.

#### 3.1.4. Effects on Osteoblast/Osteocyte Anabolism and Matrix Quality

Through ERβ, estrogen supports osteoblast differentiation and matrix production and may indirectly favor Wnt/β-catenin activity by curbing antagonists induced during high-turnover states. In osteocytes, ER-mediated signaling reduces apoptosis and helps maintain the lacuno-canalicular network, buffering increases in sclerostin and other catabolic signals that accompany estrogen loss [26].

#### 3.1.5. Functional Selectivity of S-Equol at ERβ

S-equol binds ERβ with higher affinity and transactivation potency than ERα, providing functional selectivity toward ERβ-driven programs in bone [10,11]. This pharmacology aligns with preclinical observations of reduced osteoclastogenesis (via OPG upregulation/RANKL downregulation and cytokine suppression), preservation of osteoblast function, and maintenance of trabecular structure under S-equol exposure [12,27]. Because ERβ-dominant signaling can achieve anti-resorptive and anti-oxidative effects without robust ERα activation, it offers a biologically plausible route to skeletal benefit with a more favorable systemic risk profile than nonselective estrogen therapies [10,25].

#### 3.1.6. Safety Context

Systemic estrogen therapy in the WHI estrogen-alone and estrogen plus progestin trial reduced fractures but increased stroke and venous thromboembolism [15,30]. Although WHI did not test ER subtype–selective ligands, these outcomes highlight the appeal of ERβ-preferential strategies such as S-equol that aim to retain skeletal protection while minimizing ERα-linked risks [13,27]. A recent secondary analysis of the WHI further showed that hormone therapy did not increase cardiovascular disease risk in women aged 50–59 years with vasomotor symptoms but was associated with a higher risk in those aged ≥70 years [31], reinforcing the need for safer, age-appropriate alternatives such as ERβ-selective compounds. This is especially important given the greater risk of osteoporotic fractures in those over age 70.

In summary, ERβ signaling in bone (i) restrains osteoclastogenesis through OPG/RANKL modulation and inflammatory-pathway suppression; (ii) protects osteoblasts and osteocytes through anti-oxidative, pro-survival pathways; and (iii) preferentially stabilizes trabecular integrity where remodeling surfaces are greatest [12,26,27]. S-equol’s ERβ-selective action provides a mechanistic basis for leveraging estrogen’s skeletal benefits with potentially improved safety compared with broad estrogen replacement [13,27].

### 3.2. Soy Isoflavones and the Microbiome–S-Equol Axis

#### 3.2.1. Population Prevalence and Interindividual Variability in S-Equol Production

Daidzein, a principal soy isoflavone, can be transformed by select intestinal bacteria into S-equol, a metabolite with greater biological activity than its precursor [10]. The capacity to produce S-equol varies by population and individual. Approximately 20–30% of adults in Western cohorts and 50–70% in many Asian cohorts are equol producers [11]. This population contrast likely reflects long-term dietary patterns and distinct gut microbial ecologies that favor (or fail to favor) the daidzein to S-equol pathway (Figure 1) [32].

#### 3.2.2. From Daidzin to S-Equol: Pathway Steps and Points of Control in Equol Producers

Once ingested, soy isoflavones follow a defined microbial route: the glycoside (daidzin) is hydrolyzed to daidzein, which is then sequentially reduced to dihydrodaidzein and tetrahydrodaidzein before stereoselective formation of S-equol (Figure 1) [32]. Genome-mining and pathway reconstruction analyses indicate that the full reductive toolkit is concentrated in a limited set of members of the normal gut microbiota, which helps explain why equol production is restricted to a subset of hosts [33,34]. Beyond the presence of these organisms, pathway expression appears sensitive to the surrounding community and diet, e.g., regular soy intake (substrate supply), fermentable fiber (short-chain fatty acids and pH), and cross-feeding partners that supply reducing equivalents all increase the likelihood and extent of conversion [34,35]. Conversely, factors that disrupt community structure (e.g., antibiotics) or shift luminal redox and pH can suppress conversion even when S-equol-capable taxa are present [35]. These observations suggest practical levers, e.g., dietary patterning, pre/probiotic strategies, or targeted consortia, that could enhance daidzein to S-equol flux in non-producers and stabilize production in intermittent producers [34,35].

### 3.3. Experimental and Clinical Evidence

#### 3.3.1. Animal Models

Across estrogen-deficient rodent models, S-equol reliably prevents trabecular bone loss, improves microarchitecture, and enhances biomechanical strength, with typical increases in trabecular bone volume and number of 10–20% relative to ovariectomized controls [16,36,37]. In ovariectomized mice and rats, S-equol preserves trabecular bone volume fraction, trabecular number, and connectivity and attenuates increases in trabecular spacing, with several studies demonstrating partial restoration of femoral BMD toward sham levels [16,17,27]. These structural benefits are accompanied by suppressed osteoclastogenesis, reflected in reduced RANKL and cathepsin K expression and increased OPG [16,17]. Equol also activates antioxidant and prosurvival signaling in osteoblast-lineage cells, including PI3K/Akt and Nrf2 pathways [27,38]. Several studies additionally show reductions in marrow inflammatory cytokines and adipogenic signaling (e.g., IL-6, TNF-related genes, PPARγ), consistent with attenuation of OVX-induced activation of the bone–immune–fat axis [16,17,27,38].

#### 3.3.2. Human RCTs of Soy Isoflavones and S-Equol on Bone

Because soy isoflavones exhibit estrogen receptor activity, many RCTs have tested their skeletal effects in postmenopausal women using mixed isoflavone extracts or single agents such as genistein [39,40,41,42]. Results are inconsistent, and quantitative syntheses generally report at most modest, typically at the lumbar spine, with substantial heterogeneity by dose, duration, region, ethnicity, and baseline bone status [9,43,44]. Very few RCTs have evaluated the effects of soy isoflavones on bone in men.

Several RCTs have conducted secondary analyses to examine whether equol-producer status modifies the skeletal effects of soy isoflavones (Table 2). In a 12-month RCT of 54 early postmenopausal Japanese women prospectively classified by urinary equol phenotype, equol producers lost significantly less total hip BMD than non-producers, whereas no difference was observed in placebo recipients [45]. An 8-week crossover RCT in 58 Japanese women going through menopause similarly reported that urinary deoxypyridinoline, a marker of bone resorption, decreased significantly by >30% only among equol producers [46]. In a 2-year study of 89 European postmenopausal women with or at risk for osteoporosis, soymilk supplementation alone prevented lumbar spine bone loss, and a nonsignificant trend toward greater BMD gain in equol producers (+2.8%) compared with non-producers (+0.6%) [47]. In contrast, two large Western RCTs, one in 202 Dutch postmenopausal women [48] and another in 248 early postmenopausal U.S. women [42], reported no differences in BMD or bone turnover markers by equol phenotype, despite 27–30% of participants being equol producers. Neither study demonstrated an overall beneficial effect of soy isoflavone supplementation on bone health, with evaluation by equol-producing status conducted only as secondary analysis.

Among trials testing S-equol directly, two illustrate both promise and current constraints. A 12-month RCT among 93 postmenopausal non-equol-producing Japanese women, 10 mg/day (but not 2, or 6 mg/day) reduced urinary deoxypyridinoline by 24% and attenuated whole-body DXA BMD loss [19]. In a separate 12-month RCT among 60 postmenopausal women, a combination of 10 mg S-equol plus 25 mg resveratrol improved bone turnover biomarkers (deoxypyridinoline, osteocalcin, bone-specific alkaline phosphatase) and increased whole-body DXA BMD as compared to placebo [20].

#### 3.3.3. Recommended Dose of S-Equol for Human RCTs

Pharmacokinetic data demonstrate that oral administration of 10 mg S-equol results in rapid absorption, reaching peak plasma concentrations within one hour and maintaining measurable levels (~20 ng/mL) after 24 h, consistent with an elimination half-life of approximately eight hours [49]. Across available RCTs, 10 mg/day has been the most frequently used and consistently effective dose, improving outcomes in menopausal symptoms [50,51,52], arterial stiffness [53], skin aging [54], and bone health [19]. Comparative trials indicate that 10 mg/day yields similar efficacy to higher doses (20–40 mg/day) [51] and greater benefits than lower doses (2–6 mg/day) [19]. Together, these findings support 10 mg/day as an evidence-based, physiologically relevant, and safe dose for evaluating S-equol’s skeletal effects in human trials.

### 3.4. HR-pQCT: Microarchitecture and Strength Surrogates Linked to Fracture

DXA assesses BMD but cannot differentiate cortical from trabecular compartments or detect microstructural deterioration, whereas newer modalities such as HR-pQCT allow detailed characterization of bone microarchitecture and strength [23]. Given S-equol’s hypothesized predominance in trabecular envelopes, outcomes that resolve compartment-specific microarchitecture are essential. HR-pQCT quantifies volumetric (v)BMD, geometry, and microarchitecture at the distal and diaphyseal tibia and radius, separately in trabecular and cortical compartments [23]. HR-pQCT images can be used to estimate whole-bone strength via micro–finite element analysis, which decomposes the 3D voxel model into elements to compute element-wise stresses, strains, and failure. The estimated failure load integrates the distribution of mineralized tissue, global geometry, compartmental densities, cortical porosity, and trabecular network topology (Figure 2).

HR-pQCT parameters predict fractures in both men [55] and women [56,57,58]. The Bone Micro-architecture International Consortium combined data from 6 cohorts to conduct a prospective study of 4768 women and 1995 men (mean age, 68 years; range, 40–96) [59]. Cortical and trabecular bone density and microarchitecture predicted fracture risk, independent of areal BMD (aBMD) and the Fracture Risk Assessment tool or FRAX. In the Osteoporotic Fractures in Men Study (MrOS), a large, multicenter, prospective cohort of community-dwelling older men in the U.S., the hazard ratio (HR) for all clinical fractures for each standard deviation (SD) lower failure load was 1.8 (95% confidence interval (CI): 1.4, 2.4) at the distal radius and 1.4 (95% CI: 1.1, 1.9) at the distal tibia, independent of aBMD and FRAX^®^ [55]. Trabecular and cortical vBMD, trabecular number, trabecular and cortical thickness, trabecular and cortical bone area were also independently related to fractures in MrOS. A systematic review of 40 studies showed both radial and tibial HR-pQCT parameters were significantly lower in fracture subjects with differences ranging from −2.6% (95% CI: −3.4, −1.9) in radial cortical vBMD to −12.6% (95% CI: −15.0, −10.3) in radial trabecular vBMD [60]. Thus, HR-pQCT parameters predict fracture in both men and women, and offer additional information on fracture risk beyond aBMD and clinical risk factors. These imaging advances provide a powerful framework to evaluate S-equol’s potential effects on trabecular and cortical bone compartments in future studies.

## 4. Discussion

Across mechanistic, translational, and clinical studies, a consistent pattern emerges: S-equol appears to influence bone remodeling in ways that are biologically plausible [12,27] and partially supported by early human evidence [19,20]. The ERβ-biased actions observed in preclinical models [16,17,35,36,37,38,61] align with the direction of changes reported in human trials, particularly reductions in bone resorption at 10 mg/day. Although these effects are modest and short-term, they align with microarchitectural preservation demonstrated in animal studies and suggest a coherent signal across evidence levels. Variability in clinical outcomes also mirrors differences in equol exposure [45,46,47], underscoring the importance of producer phenotype and gut microbiome composition. These observations support the hypothesis that S-equol may represent a microbiome-dependent approach to skeletal aging, while highlighting the need for larger and longer-duration trials to clarify its clinical relevance.

The broader soy-isoflavone RCT literature in postmenopausal women has yielded mixed results with at most modest average effects on spine BMD and substantial heterogeneity by dose, duration, region, ethnicity, and baseline bone status [9,11,43,44]. However, most human trials have been limited by small sample sizes, short intervention durations, reliance on DXA rather than compartment-specific imaging, and the absence of phenotyping for equol producers, all of which reduce sensitivity to detect true skeletal effects. A key explanatory factor is differential exposure to the active metabolite: only a subset of individuals produce S-equol after daidzein intake, and most trials neither stratified by nor adjusted for equol-producer status [45]. Trials and secondary analyses that did consider phenotype generally observed stronger skeletal signals among producers, consistent with a causal role for S-equol rather than for parent isoflavones per se [45,46,47]. Recent high-level evidence from an umbrella review of 10 meta-analyses [43] further contextualizes these findings, showing that soy isoflavone interventions yield modest but statistically significant increases in BMD, about 1.9% at the lumbar spine, 2.0% at the femoral neck, and 0.3% at the total hip, highlighting the biological signal that may be amplified among equol producers.

S-equol’s preferential activation of ERβ provides a coherent mechanistic basis for its potential skeletal effects [12]. ERβ signaling suppresses osteoclastogenesis by regulating OPG/RANKL and attenuating the inflammatory pathway, while supporting osteoblast and osteocyte survival under oxidative stress [17,27,38]. These actions are particularly relevant to trabecular-rich compartments where remodeling surfaces are abundant. These mechanistic pathways align with findings from human trials reporting a reduction in bone resorption markers and modest shifts toward improved turnover balance [19,20]. In parallel, the capacity to produce S-equol depends on specific gut microbiome taxa and is influenced by diet and community structure, creating inter-individual differences in internal exposure that may contribute to the heterogeneous outcomes observed across clinical studies [32,35].

Within the broader context of soy-derived phytoestrogens, genistein and daidzein have long been recognized for their beneficial effects on bone remodeling, supported by clinical trials showing improvements in bone turnover and BMD, particularly with 54 mg/day genistein aglycone [62]. However, genistein binds both estrogen receptors and exhibits relatively higher ERα affinity than S-equol, raising theoretical concerns about proliferative effects at breast and uterine tissues despite overall reassuring trial data [11]. Daidzein itself has weak estrogen-receptor binding and exerts limited biological activity unless converted to S-equol by specific gut microbial taxa. S-equol, in contrast, is a high-affinity, highly selective ERβ agonist, a profile associated with anti-proliferative effects in reproductive tissues and potentially greater skeletal specificity [11]. Because equol production varies widely across individuals, direct S-equol supplementation offers standardized exposure and avoids dependence on microbial conversion, which may reduce heterogeneity in clinical responses compared with whole-soy foods or mixed-isoflavone preparations. These distinctions highlight why S-equol may represent a more targeted and potentially safer phytoestrogenic strategy for skeletal aging, although head-to-head comparative trials remain needed.

Human evidence remains constrained by several fundamental gaps. First, most trials do not phenotype equol producer status in advance, making it difficult to interpret heterogeneous responses and limiting causal inference [42,45]. Second, nearly all studies rely on DXA rather than compartment-specific imaging such as HR-pQCT, which is necessary to assess cortical and trabecular microstructure [27]. Third, men remain markedly underrepresented, restricting generalizability beyond postmenopausal women. Other constraints, including small sample sizes, short durations, variations in dosing or formulation (S-equol versus isoflavone mixtures), and inconsistently captured adherence or background diet, further contribute to uncertainty.

Background dietary patterns may also contribute to variability in isoflavone trial outcomes, as Mediterranean, plant-based, or Nordic diets provide polyphenols, fiber, and anti-inflammatory nutrients that influence bone metabolism [63,64]. Although S-equol trials typically use standardized supplements, differences in habitual diet may still modify responses and warrant consideration in future research. Habitual physical activity is another determinant of bone turnover, yet most existing trials do not systematically measure or control exercise patterns. These factors may contribute to variability in observed effects and should be more consistently incorporated into future trial designs.

A closer examination of trials of soy isoflavones stratifying by equol-producing status (Table 2) shows that inconsistent findings likely reflect methodological heterogeneity rather than true biological contradictions. Most trials did not report the specific daidzein/genistein composition of the isoflavone supplements, making it difficult to compare interventions across studies or interpret how differences in supplement composition may have influenced the observed outcomes. Adherence assessment also varied: some studies monitored compliance via pill counts, while others did not report adherence procedures in the available text. In addition, studies differed in key population characteristics, including age, years since menopause, habitual soy intake, and underlying prevalence of equol producers, all factors known to influence S-equol formation. Definitions of equol-producer status also varied across trials, including serum thresholds and urinary ratios, contributing further inconsistency. These variations help explain why some studies observed benefits among equol producers while others did not, despite similar overall isoflavone doses. The conflicting results summarized in Table 2 most plausibly arise from differences in phenotype ascertainment, population characteristics, adherence, and endpoint sensitivity, rather than a fundamental inconsistency in the underlying biology of S-equol.

S-equol’s safety profile is supported by its preferential activation of ERβ and its minimal affinity for ERα, which reduces concerns about uterine, breast, and thrombotic risks. However, a more cautious interpretation is warranted. Most available safety data derive from short- to medium-term trials lasting weeks to months, and long-term endocrine, cardiovascular, and oncologic safety remains insufficiently characterized. Although no serious adverse events have been attributed to S-equol supplementation in published studies, comprehensive safety monitoring, including vascular, hepatic, and reproductive endpoints, will be essential in future trials, particularly those of longer duration or involving diverse populations. Thus, while S-equol’s mechanistic selectivity suggests a favorable theoretical profile, definitive conclusions about long-term safety cannot yet be drawn.

Beyond S-equol itself, broader lifestyle and nutritional factors may also influence skeletal and cardiometabolic health in postmenopausal women. Mediterranean-style dietary patterns, which are rich in polyphenols, fiber, and anti-inflammatory nutrients, have demonstrated favorable effects on cardiovascular disease, metabolic risk factors, and fracture risk, and may complement the actions of soy isoflavones [65]. Although S-equol was the primary focus of this review, combined approaches incorporating healthy dietary patterns, other nutraceuticals, and regular physical activity may yield additive benefits for noncommunicable disease prevention. Future studies should consider these multidimensional lifestyle factors when evaluating the role of phytoestrogen supplementation.

DXA cannot detect compartment-specific or microarchitectural changes that are central to fragility risk. HR-pQCT enables in vivo assessment of volumetric BMD, trabecular number/thickness/separation, cortical thickness/porosity, and micro–finite element-derived estimates of stiffness [23]. Importantly, HR-pQCT provides clinically meaningful information by predicting incident fractures independently of DXA and FRAX and by identifying microarchitectural deterioration in individuals who do not meet DXA criteria for osteoporosis. Large multicenter studies, including the Bone Microarchitecture International Consortium, demonstrate that HR-pQCT–derived measures such as trabecular density and estimated failure load nearly double fracture risk per SD decrement, underscoring its prognostic value [66]. Given the hypothesized trabecular predominance of ERβ-mediated benefit, HR-pQCT provides a more sensitive and pathophysiologically aligned endpoint set than DXA alone and should be incorporated alongside standard bone turnover markers.

Overall, the combined evidence supports a coherent explanatory model in which S-equol’s ERβ-selective actions provide strong biological plausibility for skeletal benefit, while microbiome-dependent variation in equol production and methodological constraints in human trials modulate the observable effect size. The consistency of preclinical findings reflects tightly controlled exposure and reliable engagement of ERβ pathways, whereas human outcomes vary because internal S-equol levels depend on producer phenotype, supplement composition, and habitual diet—factors rarely standardized across trials.

Future trials should standardize phenotype ascertainment using harmonized soy challenge test protocols and validated plasma or urine thresholds, with repeat testing to capture temporal stability. Because producer status may be modifiable (through substrate supply, fermentable fiber, or microbiome-directed strategies), trials should predefine how such factors are monitored or controlled. Reporting should include exposure metrics (circulating/urinary S-equol) to link phenotypic and pharmacokinetic factors to skeletal outcomes. Although full standardized challenge testing may be difficult to implement in large multicenter trials, simplified approaches could offer practical alternatives, such as deep metagenomic sequencing of the stool microbiome to infer equol-production capacity [67].

S-equol’s ERβ-preferential profile offers a theoretical safety advantage relative to nonselective estrogen therapy. Nonetheless, longer trials should include systematic monitoring of vascular events, endometrial outcomes (where applicable), thyroid and hepatic function, and drug–nutrient interactions. Available short-term data have not signaled major safety concerns at commonly studied doses [19,50,51,52,53,54], but fracture-relevant durations and adequate sample sizes will be needed to confirm the benefit–risk balance.

Future research should clarify the long-term skeletal relevance of S-equol by addressing several gaps. Important future directions include generating larger and longer-duration human evidence, incorporating advanced imaging and relevant biomarker panels to better characterize bone microstructure and mechanistic pathways, and examining outcomes in both women and men across midlife and older age. Comparative studies evaluating S-equol alongside broader isoflavone mixtures, as well as investigations into microbiome-related factors that influence equol production, would further help define where S-equol may offer unique advantages.

## 5. Conclusions

Current evidence suggests that S-equol may hold potential as a microbiome-dependent, ERβ-biased approach to supporting skeletal health, but definitive conclusions remain limited by the small number and short duration of existing human trials. Evidence from clinical studies suggests that 10 mg/day of S-equol is a physiologically relevant and safe dose capable of improving bone turnover and preserving BMD in short-term trials, although longer-term studies are needed to confirm sustained efficacy and safety. Integrating microbiome phenotyping, HR-pQCT imaging, and biomarker profiling will enable future trials to clarify efficacy and mechanisms of S-equol. S-equol represents a promising model for precision nutrition, where microbial, hormonal, and host factors converge with the potential to prevent age-related bone fragility. Rigorous trials integrating microbiome phenotyping and advanced imaging are essential to validate this approach and translate the mechanistic promise into clinical benefit.

## Figures and Tables

**Figure 1 nutrients-17-03962-f001:**
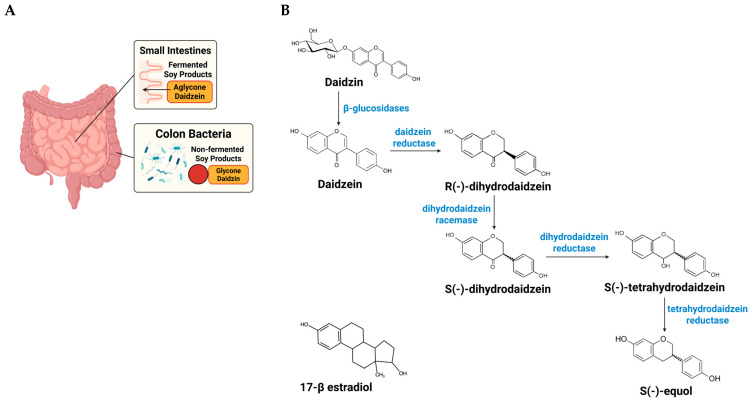
Site-dependent absorption of daidzin and host–microbe pathway from dietary daidzin to S-equol. The schematic illustrates the sequential conversion of soy isoflavones and the locations where these steps occur along the gut. (**A**) Anatomical/absorption context: in fermented soy foods (e.g., tempeh, miso), isoflavones are largely in aglycone form (daidzein), which is preferentially absorbed in the small intestine, leaving less substrate for colonic metabolism. In non-fermented soy foods (e.g., tofu, soy milk), isoflavones are predominantly glycosides (daidzin) that reach the colon, where hydrolysis and the reductive steps above can generate S-equol in individuals harboring the requisite bacteria. (**B**) Chemical pathway (colon): the glycoside daidzin is hydrolyzed by intestinal/microbial β-glucosidases to the aglycone daidzein; selected resident gut bacteria then catalyze a series of anaerobic reductions: daidzein reductase converts daidzein to R-(−)-dihydrodaidzein; dihydrodaidzein racemase interconverts R-(−)-dihydrodaidzein to S-(−)-dihydrodaidzein; dihydrodaidzein reductase reduces S-(−)-dihydrodaidzein to S-tetrahydrodaidzein; and tetrahydrodaidzein reductase converts S-tetrahydrodaidzein to S-(−)-equol. Because the full enzyme set resides in a limited subset of the normal gut microbiota, only some individuals are equol producers, and production can vary with diet and microbial community context. “The structure of 17β-estradiol is shown to demonstrate its similarity to S-equol. The figure was created using BioRender (https://www.biorender.com) under publication license.

**Figure 2 nutrients-17-03962-f002:**
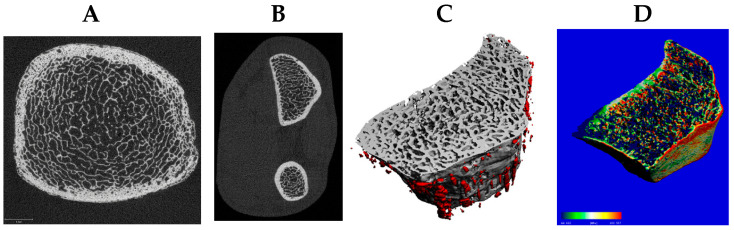
The distal tibia (**A**) and radius (**B**) are standard sites for high-resolution peripheral quantitative computed tomography (HR-pQCT) (61 µm resolution) to assess trabecular and cortical microarchitecture, including cortical porosity of the radius (**C**) and bone strength via micro-finite element analysis (**D**).

**Table 1 nutrients-17-03962-t001:** Comparative roles of estrogen receptor αs and β, and their agonists, in bone cell function and bone metabolism.

	**Estrogen Receptor** **α**	**Estrogen Receptor** **β**
Cellular expression [12]	Osteoblasts, osteoclasts, osteocytes	Osteoblasts, osteoclasts, chondrocytes
Bone type [12]	Predominantly in cortical bone	Predominantly in trabecular bone
Effect on osteoblasts [12]	Increases proliferation and differentiation	Modulates differentiation and mineralization
Effect on osteoclasts [12,13]	Strong inhibition	Moderate inhibition
Role in bone mass regulation [12,13]	Dominant in maintaining bone density	Modulatory role in bone turnover
Expression with age [28]	Declines significantly	Relatively stable
	**Estrogen receptor** **α agonist**	**Estrogen receptor** **β agonist**
Primary Role in Bone Health [17,26,28]	Major regulator of bone formation and resorption	Modulator of bone metabolism and inflammation
Effect on Osteoblasts (Bone Formation) [27]	Stimulates osteoblast differentiation and activity, leading to increased bone formation	Minimal or inhibitory effect on osteoblasts
Effect on Osteoclasts (Bone Resorption) [27]	Strong inhibition of osteoclast differentiation, reducing bone loss	Moderate inhibition of osteoclast activity, but less potent than estrogen receptor α
Impact on BMD [29]	Increases BMD and prevents bone loss	May help preserve BMD, but much weaker than estrogen receptor α agonist
Effect on Bone Turnover [27]	Reduces bone turnover, maintaining bone homeostasis	May slightly lower bone turnover, but effects are less clear
Mechanisms Involved [27]	Inhibits RANKL-induced osteoclastogenesis -Stimulates Wnt/β-catenin signaling for osteoblast function	Modulates inflammation in bone May balance estrogen receptor α activity to fine-tune bone remodeling
Clinical Use in Osteoporosis	Used in hormone therapy and SERMs (e.g., raloxifene) for osteoporosis treatment	Not widely used for osteoporosis treatment alone

Note: Shaded rows are used to distinguish receptor-level characteristics from agonist-specific effects for clarity. BMD: Bone mineral density, RANKL: Receptor Activator of Nuclear Factor-κB Ligand, SERM: Selective estrogen receptor modulator.

**Table 2 nutrients-17-03962-t002:** Randomized controlled trials that examined soy isoflavones and bone outcomes stratified by equol-producer status (secondary analyses).

Study, Year, Reference	Population	Intervention (Dose & Duration)	% Equol Producers and Equol-Producer Analysis	Findings on Bone and Bone Markers
Uesugi et al. 2004 [46]	58 Japanese climacteric women	40 mg/day soy isoflavones for 8 weeks (crossover RCT)	40%; Urinary equol >0.01 nM/mM creatinine during the intervention	Urinary deoxypyridinoline decreased significantly in equol producers but not in non-producers.
Lydeking-Olsen et al. 2004 [47]	89 European postmenopausal women with or at high risk for osteoporosis	Soymilk (76 mg/day soy isoflavones) and/or progesterone for 24 months	30%; Serum equol >20 nM/L during the intervention	Soymilk alone prevented lumbar-spine bone loss vs. placebo; equol producers showed greater mean DXA BMD gain (+2.8%) than non-producers (+0.6%), not statistically significant.
Kreijkamp-Kaspers et al. 2004 [48]	202 Dutch postmenopausal women	25.6 g soy protein with 99 mg isoflavones/day vs. milk-protein placebo for 12 months	30%; Plasma equol >83 nmol/L after 12 months of intervention	Overall, no significant effect on DXA BMD, lipids, or cognition; no difference between equol producers and non-producers.
Wu et al. 2007 [45]	54 early postmenopausal Japanese women	75 mg/day isoflavone conjugates for 12 months	46%; Urinary equol ≥1.0 µmol/L after soy challenge test	Equol producers lost significantly less total-hip DXA BMD than non-producers; effect not observed in placebo group.
Levis et al. 2011 [42]	248 early postmenopausal women in the U.S.	200 mg/day soy isoflavone tablets for 24 months	27%; Urinary equol ≥1000 nM after supplementation	Overall, no differences in DXA BMD or bone turnover markers; no benefit among equol producers.

BMD: Bone Mineral Density, RCT: Randomized Controlled Trial.

## Data Availability

No new datasets were generated or analyzed. All information synthesized is from publicly available, cited sources.

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
