# Peer review of "S-Equol as a Gut-Derived Phytoestrogen Targeting Estrogen Receptor β: A Promising Bioactive Nutrient for Bone Health in Aging Women and Men: A Narrative Review"

_nutrients, 2025, doi:10.3390/nu17243962_

Round 1
Reviewer 1 Report
Comments and Suggestions for Authors
Evaluation of manuscript nutrients-4014793
The manuscript addresses a highly relevant and timely topic in the field of nutrition and bone health. The review presents a comprehensive and well-supported synthesis of the evidence on S-Equol.
The Abstract is descriptive but lacks concrete information that would allow the reader to assess the scope of the review. I recommend including the number of studies analyzed and, in the results presentation, highlighting key conclusions, such as the consistently effective dose of 10 mg/day or the patterns observed in Table 2.
The article is generally well-written and structured, logically guiding the reader from the clinical problem context through mechanisms of action. However, an important detail is missing in the introduction: How can the synthesis of these studies contribute to advancing science? Are there divergent points? How can this synthesis help form a new understanding of osteoporosis treatment?
Regarding the Methods section, it is incomplete even for a narrative review. It is essential to detail the eligibility criteria (e.g., study type, population, outcomes) and specify the databases consulted along with the search dates. The justification for choosing a narrative review over a systematic review should also be explained, whether due to the exploratory nature of the research question, methodological heterogeneity among studies, or the aim to contextualize mechanisms and gaps more broadly. If the authors gathered enough studies to construct Table 2, why not opt for a systematic review?
In terms of content, the analysis of clinical evidence could be deepened. Although studies with conflicting results are presented, a more critical discussion of potential sources of this heterogeneity, such as differences in doses, formulations, or baseline population characteristics, would add a valuable layer of interpretation. Furthermore, while the discussion of S-Equol's safety profile is well-grounded in its ERβ selectivity, a slightly more cautious tone would be beneficial. It would be advantageous to explicitly acknowledge that most safety data come from short- and medium-term studies and to emphasize the importance of robust safety monitoring and long-term studies to confirm the promising theoretical profile.
Additionally, minor adjustments, such as correcting typographical errors (e.g., "Toy isoflavones" in the abstract), enriching the caption for Figure 2, and providing more detailed methodological criteria for literature search and selection, would further strengthen the manuscript.
The current Discussion is limited to summarizing results and lacks critical analysis and expert opinion. As researchers in the field, the authors should integrate their personal interpretation of the findings, offering perspectives on contradictions in the literature, the weight of the evidence, and the practical implications of the discoveries, moving beyond mere description to propose a well-founded academic position.
Author Response
Comment:
The manuscript addresses a highly relevant and timely topic in the field of nutrition and bone health. The review presents a comprehensive and well-supported synthesis of the evidence on S-Equol.
We appreciate thoughtful and constructive feedback, which has significantly improved the clarity, rigor, and balance of the manuscript. All comments have been addressed in the revised version as follows.
- The Abstract is descriptive but lacks concrete information that would allow the reader to assess the scope of the review. I recommend including the number of studies analyzed and, in the results presentation, highlighting key conclusions, such as the consistently effective dose of 10 mg/day or the patterns observed in Table 2.
Response:
We agree and revised the Abstract to include specific quantitative findings and conclusions.
Revised text
A limited number of human trials show reductions in bone resorption by 20% at a daily dose of 10 mg S-equol.
We clarified that findings in Table 2 are not uniformly consistent across equol-producer groups and addressed this in the Discussion.
Comment:
- The article is generally well-written and structured, logically guiding the reader from the clinical problem context through mechanisms of action. However, an important detail is missing in the introduction: How can the synthesis of these studies contribute to advancing science? Are there divergent points? How can this synthesis help form a new understanding of osteoporosis treatment?
Response:
The Introduction now explicitly describes the scientific contribution and divergent findings.
Revised text (excerpt)
Despite abundant research across mechanistic, microbiome, and clinical domains, these bodies of evidence have rarely been integrated to explain why responses to isoflavones and S-equol differ across individuals and populations. This synthesis is needed to clarify how biological selectivity, phenotypic variability, and measurement limitations jointly shape observed outcomes.
These divergent findings underscore the importance of synthesizing evidence across mechanistic, microbiome, and imaging domains to clarify when, why, and for whom S-equol is likely to confer skeletal benefit.
Comment:
- Regarding the Methods section, it is incomplete even for a narrative review. It is essential to detail the eligibility criteria (e.g., study type, population, outcomes) and specify the databases consulted along with the search dates. The justification for choosing a narrative review over a systematic review should also be explained, whether due to the exploratory nature of the research question, methodological heterogeneity among studies, or the aim to contextualize mechanisms and gaps more broadly. If the authors gathered enough studies to construct Table 2, why not opt for a systematic review?
Response:
Methods were substantially expanded to include eligibility criteria, search strategy, timeframe, databases, and explicit rationale for narrative synthesis.
Revised text (excerpt)
We included randomized controlled trials (RCTs), prospective and cross-sectional observational studies, animal models of estrogen deficiency, and in vitro mechanistic studies that examined S-equol, soy isoflavones, or related agents affecting ERβ signaling or microbial equol production. Eligible human studies enrolled postmenopausal women or men, while eligible preclinical studies used ovariectomized or otherwise estrogen-deficient rodent models… A narrative synthesis was selected for three reasons. First, the research question spans mechanistic, translational, and human clinical evidence, requiring integrated interpretation that cannot be captured by quantitative pooling alone...
Comment:
- In terms of content, the analysis of clinical evidence could be deepened. Although studies with conflicting results are presented, a more critical discussion of potential sources of this heterogeneity, such as differences in doses, formulations, or baseline population characteristics, would add a valuable layer of interpretation. Furthermore, while the discussion of S-Equol's safety profile is well-grounded in its ERβ selectivity, a slightly more cautious tone would be beneficial. It would be advantageous to explicitly acknowledge that most safety data come from short- and medium-term studies and to emphasize the importance of robust safety monitoring and long-term studies to confirm the promising theoretical profile.
Response:
A new paragraph critically examines sources of heterogeneity. A new safety section adds balanced interpretation
Revised text (excerpt)
A closer examination of trials of soy isoflavones stratifying by equol-producing status (Table 2) shows that inconsistent findings likely reflect methodological heterogeneity rather than true biological contradictions. Most trials did not report the specific daidzein/genistein composition of the isoflavone supplements, making it difficult to compare interventions across studies or interpret how differences in supplement composition may have influenced the observed outcomes. Adherence assessment also varied: some studies monitored compliance via pill counts, while others did not report adherence procedures in the available text. In addition, studies differed in key population characteristics, including age, years since menopause, habitual soy intake, and underlying prevalence of equol producers, all factors known to influence S-equol formation…
As for the safety, we now emphasize that while the ERβ-selective mechanism suggests favorable safety, longer-term trials with systematic monitoring of adverse events are essential to confirm the theoretical safety profile in diverse populations. We added the following paragraph in the discussion.
Revised text
S-equol’s safety profile is supported by its preferential activation of ERβ and minimal affinity for ERα, which reduces concerns related to uterine, breast, and thrombotic risk. However, a more cautious interpretation is warranted. Most available safety data derive from short- to medium-term trials lasting weeks to months, and long-term endocrine, cardiovascular, and oncologic safety remains insufficiently characterized. Although no serious adverse events have been attributed to S-equol supplementation in published studies, comprehensive safety monitoring—including vascular, hepatic, and reproductive endpoints—will be essential in future trials, particularly those of longer duration or involving diverse populations. Thus, while S-equol’s mechanistic selectivity suggests a favorable theoretical profile, definitive conclusions about long-term safety cannot yet be drawn.
Comment:
- Additionally, minor adjustments, such as correcting typographical errors (e.g., "Toy isoflavones" in the abstract), enriching the caption for Figure 2, and providing more detailed methodological criteria for literature search and selection, would further strengthen the manuscript.
Response:
We carefully reviewed the entire manuscript and abstract, but were unable to locate the typographical error “toy isoflavones.” We confirmed that all instances of “soy isoflavones” are spelled correctly. Nonetheless, we rechecked the manuscript thoroughly for any additional typographical issues and corrected minor formatting inconsistencies.
Comment:
- The current Discussion is limited to summarizing results and lacks critical analysis and expert opinion. As researchers in the field, the authors should integrate their personal interpretation of the findings, offering perspectives on contradictions in the literature, the weight of the evidence, and the practical implications of the discoveries, moving beyond mere description to propose a well-founded academic position.
Response:
We greatly expanded the Discussion with a unifying interpretation of mechanistic, microbiome, and clinical findings.
Revised text
Overall, the combined evidence supports a coherent explanatory model in which S-equol’s ERβ-selective actions provide strong biological plausibility for skeletal benefit, while microbiome-dependent variation in equol production and methodological constraints in human trials modulate the observable effect size. The consistency of preclinical findings reflects tightly controlled exposure and reliable engagement of ERβ pathways, whereas human outcomes vary because internal S-equol levels depend on producer phenotype, supplement composition, and habitual diet—factors rarely standardized across trials.
Reviewer 2 Report
Comments and Suggestions for Authors
nutrients-4014793-peer-review-v1
The manuscript presents interesting plan for a review paper, but the problem is that authors have presented a manuscript with not sufficient information. The work looks like it is prepared for a very specialized audience where all topics are presented as much as possible in a reduced way. In my opinion paper needs extensive revision where all (almost all) topics needs to be extended and presented with more details; authors need to provide more details, more examples, to be more critical (where possible), looking for alternatives. In fact, the point is the review paper will serve not only limited number of very specialized colleagues but needs to be targeted at a broader audience, and related to this, authors will need to provide more information for all parts in the manuscript.
Affiliations needs to be presented with complete details. Name of the city and country needs to be provided.
Ln49: Please, remove additional full stop between text and reference. Please, check entire manuscript for similar adjustments. The topic presented under 3.1.1. is very short and in my opinion deserve better attention form the authors. In current way, based only on 2 sentences is very preliminary. Authors will need to extend the topic, present more results in the text, even if this was already mentioned as Table 1. Simply present the most important data from Table 1 summary as textual material in this place.
In fact, similar point is related to the information presented in all subtopics further presented till 3.1.6.
Same point again. Under the topic 3.3.1. you have referend to 9 papers, but in same time topic related to animal models was literally limited to 3 sentences. Please, provide more details, more examples, even go to positive and negative comments in the mentioned papers and present your opinion on the topic.
Discussion needs to be updated and upgraded. Authors will need to be more original in the discussion section and clearly show their opinion in the evaluated topic.
Reference list need to be formatted according to the exigences of the Publisher and the Journal.
Author Response
Comment:
The manuscript presents interesting plan for a review paper, but the problem is that authors have presented a manuscript with not sufficient information. The work looks like it is prepared for a very specialized audience where all topics are presented as much as possible in a reduced way. In my opinion paper needs extensive revision where all (almost all) topics needs to be extended and presented with more details; authors need to provide more details, more examples, to be more critical (where possible), looking for alternatives. In fact, the point is the review paper will serve not only limited number of very specialized colleagues but needs to be targeted at a broader audience, and related to this, authors will need to provide more information for all parts in the manuscript.
Response:
We thank the reviewer for this feedback and have undertaken extensive revision to make the manuscript more accessible to a broader audience while maintaining scientific rigor. We have expanded all major sections with additional details, specific examples from the literature, critical analysis of conflicting findings, and discussion of alternative approaches where applicable.
First, we significantly expanded the mechanistic section to provide clearer explanations of ERβ-selective signaling, its downstream pathways (e.g., OPG/RANKL, oxidative stress and Nrf2 activation), and how these mechanisms relate to trabecular versus cortical bone remodeling. We also included more detailed examples from preclinical models to illustrate S-equol’s effects on bone microarchitecture and strength.
Second, we strengthened the clinical evidence section by adding more detailed descriptions of randomized trials, population characteristics, supplement formulations, adherence considerations, and phenotype definitions. We now provide critical examination of inconsistent findings—including methodological variation, differences in phenotype ascertainment, supplement composition, and trial populations—which helps reconcile contradictory results in the literature.
Third, we added a dedicated, cautious safety section addressing the limitations of short-term data, the need for long-term endocrine and cardiovascular monitoring, and the theoretical basis for S-equol’s favorable profile. This directly responds to the reviewer’s request for more complete and balanced evaluation.
Fourth, we incorporated a new integrative synthesis paragraph in the Discussion that articulates a unified conceptual framework linking mechanistic evidence, microbiome-dependent equol production, DXA limitations, and clinical outcomes. This section explicitly presents our expert interpretation of the evidence, rather than a descriptive summary.
Fifth, the Introduction and Discussion have both been revised to provide more context, better transitions, expanded explanations, and clearer justification for the narrative review approach. We also strengthened the presentation of alternatives and remaining uncertainties.
Comment: Affiliations needs to be presented with complete details. Name of the city and country needs to be provided.
Response:
All affiliations have been updated to include complete institutional details with city and country information.
Comment: Ln49: Please, remove additional full stop between text and reference. Please, check entire manuscript for similar adjustments.
Response:
We have corrected this formatting error and carefully reviewed the entire manuscript to ensure consistent and proper reference formatting throughout.
Comment: The topic presented under 3.1.1. is very short and in my opinion deserve better attention form the authors. In current way, based only on 2 sentences is very preliminary. Authors will need to extend the topic, present more results in the text, even if this was already mentioned as Table 1. Simply present the most important data from Table 1 summary as textual material in this place. In fact, similar point is related to the information presented in all subtopics further presented till 3.1.6.
Response:
We substantially expanded Section 3.1.1 by adding a more detailed explanation of ERα and ERβ compartmental distribution, cellular localization, and their functional implications for skeletal remodeling, drawing directly on the information summarized in Table 1. This revision provides a clearer context and a mechanistic rationale for receptor-specific effects. Regarding Section 3.1.6, we elected not to expand this subsection further because safety considerations are now addressed more fully in the revised Discussion.
Section 3.1.1.
Revised text
ERα and ERβ exhibit distinct spatial and cellular patterns within the skeleton that help explain their divergent functional role. ERα signaling predominates in cortical envelopes, particularly along the periosteal and endocortical surfaces [1], where it regulates cortical thickness, periosteal expansion, and endocortical resorption. Declines in ERα activity after menopause [2] contribute to cortical porosity, thinning, and loss of bending strength, which are major determinants of fracture risk. In contrast, ERβ is broadly expressed across osteoblast-lineage cells, osteocytes, and marrow stromal cells, with relatively higher representation in trabecular-rich regions where metabolic turnover is greatest [1]. These compartmental differences suggest that ERβ-selective ligands such as S-equol may exert their greatest effects within trabecular bone, where ERβ-driven anti-resorptive and cytoprotective actions are most relevant and where microarchitectural deterioration progresses rapidly after menopause. This distributional framework provides the biological rationale summarized in Table 1, linking receptor localization to compartment-specific skeletal effects [3-5].
Section 3.1.6,
Revised text
S-equol’s safety profile is supported by its preferential activation of ERβ and its minimal affinity for ERα, which reduces concerns about uterine, breast, and thrombotic risks. However, a more cautious interpretation is warranted. Most available safety data derive from short- to medium-term trials lasting weeks to months, and long-term endocrine, cardiovascular, and oncologic safety remains insufficiently characterized. Although no serious adverse events have been attributed to S-equol supplementation in published studies, comprehensive safety monitoring, including vascular, hepatic, and reproductive endpoints, will be essential in future trials, particularly those of longer duration or involving diverse populations. Thus, while S-equol’s mechanistic selectivity suggests a favorable theoretical profile, definitive conclusions about long-term safety cannot yet be drawn.
Comment: Same point again. Under the topic 3.3.1. you have referend to 9 papers, but in same time topic related to animal models was literally limited to 3 sentences. Please, provide more details, more examples, even go to positive and negative comments in the mentioned papers and present your opinion on the topic.
Response: Section 3.3.1 has been expanded as follows:
Revised text
Across estrogen-deficient rodent models, S-equol reliably prevents trabecular bone loss, improves microarchitecture, and enhances biomechanical strength, with typical increases in trabecular bone volume and number of 10–20% relative to ovariectomized controls [6-8]. In ovariectomized mice and rats, S-equol preserves trabecular bone volume fraction, trabecular number, and connectivity and attenuates increases in trabecular spacing, with several studies demonstrating partial restoration of femoral BMD toward sham levels [3, 6, 9]. These structural benefits are accompanied by suppressed osteoclastogenesis, reflected in reduced RANKL and cathepsin K expression and increased OPG [6, 9]. Equol also activates antioxidant and prosurvival signaling in osteoblast-lineage cells, including PI3K/Akt and Nrf2 pathways [3, 10]. Several studies additionally show reductions in marrow inflammatory cytokines and adipogenic signaling (e.g., IL-6, TNF-related genes, PPARγ), consistent with attenuation of OVX-induced activation of the bone–immune–fat axis [3, 6, 9, 10].
Comment: Discussion needs to be updated and upgraded. Authors will need to be more original in the discussion section and clearly show their opinion in the evaluated topic.
Response: Discussion rewritten extensively to:
- provide expert interpretation
- reconcile conflicting findings
- link mechanism with clinical outcomes
- emphasize methodological constraints
- highlight translational relevance
Comment: Reference list need to be formatted according to the exigences of the Publisher and the Journal.
Response: All references have been carefully reformatted to fully comply with the Nutrients journal style requirements.
Reviewer 3 Report
Comments and Suggestions for Authors
The authors present a manuscript entitled “S-Equol as a Gut-Derived Phytoestrogen Targeting Estrogen Receptor β: A Promising Bioactive Nutrient for Bone Health in Aging Women and Men”, which aims to synthesize mechanistic, translational, and clinical evidence on the role of S-equol in skeletal health. The topic is timely and relevant, bridging microbiome science with estrogen receptor biology in the context of osteoporosis prevention. However, several major issues remain, including overstatement of clinical efficacy from limited short-term trials, insufficient discussion of methodological constraints such as equol-producer stratification and imaging endpoints, and promotional rather than cautious phrasing in the conclusions. Addressing these concerns with clearer distinction between preclinical and human evidence, balanced interpretation, and emphasis on future research directions would substantially strengthen the manuscript’s scientific rigor and impact. See the comments below:
Abstract:
- Line 19–27 (Background/Objectives): The abstract provides a clear rationale but is overly detailed in epidemiology and background. Abstracts should be concise; the extensive description of osteoporosis burden could be shortened to emphasize the research gap.
- Line 28–30 (Methods): The methods are described vaguely. The search strategy lacks detail (timeframe, inclusion/exclusion criteria, type of studies). Without this, reproducibility and transparency are limited.
- Line 31–37 (Results): The results section mixes mechanistic, preclinical, and clinical findings but lacks quantification. Statements such as “enhances trabecular bone density” or “reduces bone turnover” are too general. Including effect sizes or trial durations would strengthen credibility.
- Line 34–35: The limitation (DXA reliance, lack of equol-producer stratification) is noted, but it should be emphasized earlier to balance enthusiasm with caution.
- Line 38–40: The mention of HR-pQCT and microbiome science is appropriate, but too technical for an abstract. This detail could be condensed into a single sentence highlighting “advanced imaging and microbiome-informed approaches.”
- Line 41–42 (Conclusions): The conclusion is promotional in tone (“exemplifies precision nutrition”). It should be reframed more cautiously, emphasizing potential rather than certainty.
Introduction:
- Line 47–57: The epidemiological context is strong, but the introduction is overly detailed with statistics. While useful, the abundance of prevalence data (U.S., global, age-specific) risks overwhelming readers. A more concise summary would improve readability.
- Line 58–65: The transition to dietary compounds is appropriate, but the narrative jumps quickly from soy isoflavones to equol without fully explaining why equol is uniquely important. The “equol-producer phenotype” is mentioned but not sufficiently contextualized in terms of clinical relevance. Please support this statement with relevant citation to the literature: 10.1111/1541-4337.70080
- Line 66–83: The mechanistic description of ERβ vs. ERα is informative, but too detailed for an introduction. The WHI discussion (Lines 75–80) is lengthy and distracts from the central focus on equol.
- Line 84–95: Preclinical and clinical evidence is presented, but the introduction mixes results with methodological critique. This section should highlight the promise of equol while reserving methodological limitations for later discussion.
- Line 96–105: The mention of DXA vs. HR-pQCT is important, but the introduction risks becoming a methods discussion. This level of detail belongs in the Discussion or Perspectives section.
- Line 106–113: The final paragraph is strong in identifying gaps, but the claim that “S-equol is now available as a dietary supplement” (Line 111–112) is promotional in tone and should be reframed more neutrally.
Discussion:
- Line 325–333: The discussion begins with a strong synthesis of mechanistic and translational data, but it is somewhat repetitive of the Introduction. The authors should focus on interpretation rather than restating background.
- Line 329–331: The human RCT evidence is presented, but trial limitations (sample size, duration, endpoints) are acknowledged only later (Line 353 onward). These limitations should be integrated earlier to balance enthusiasm with caution.
- Line 335–342: The heterogeneity in soy-isoflavone RCTs is noted, but the discussion does not adequately quantify effect sizes or provide meta-analytic context. Without this, the reader cannot gauge the magnitude of equol’s potential benefit.
- Line 343–351: Mechanistic links (ERβ signaling, microbiome axis) are well described, but speculative language (“natural experiment”) should be moderated. The discussion would benefit from integrating mechanistic insights with clinical trial outcomes more explicitly.
- Line 353–358: Limitations of human evidence are appropriately listed, but the section is dense. The authors should prioritize the most critical gaps (phenotyping, imaging, male underrepresentation) rather than listing too many secondary issues.
- Line 359–365: The emphasis on HR-pQCT is valid, but the discussion risks becoming overly technical. The authors should explain why HR-pQCT endpoints are clinically meaningful (fracture prediction, microstructural deterioration) rather than just listing parameters.
- Line 366–372: The recommendations for phenotype ascertainment are important, but the feasibility of standardized soy challenge tests in large trials should be discussed.
- Line 373–378: Safety considerations are briefly mentioned, but the discussion lacks depth. Potential endocrine, vascular, and hepatic risks should be contextualized with reference to existing phytoestrogen safety data.
- Line 379–384: The priorities listed are appropriate, but they read more like a grant proposal than a discussion. The authors should frame them as research directions rather than prescriptive trial designs.
Conclusion:
- Line 386–389: The conclusion restates mechanistic actions of S-equol (anti-resorptive, antioxidant, anti-inflammatory), but this is repetitive of earlier sections. A conclusion should synthesize findings rather than reiterate background.
- Line 390–391: The statement that “10 mg/day of S-equol is a physiologically relevant and safe dose” is too definitive. The evidence base is limited to short-term, small-scale trials. This claim should be qualified as preliminary and not presented as established fact.
- Line 392–393: The recommendation to integrate microbiome phenotyping, HR-pQCT, and biomarker profiling is appropriate, but reads more like a methods section. The conclusion should emphasize future directions rather than technical detail.
- Line 394–395: The phrase “exemplifies the emerging concept of precision nutrition” is promotional in tone. The conclusion should remain objective and highlight scientific potential while acknowledging limitations.
Comments on the Quality of English LanguageThe manuscript would benefit from professional English editing to ensure grammatical accuracy, concise expression, and consistent scientific terminology. Moderating promotional language, restructuring long sentences, and clarifying distinctions between preclinical and clinical evidence will significantly enhance clarity and readability.
Author Response
The authors present a manuscript entitled “S-Equol as a Gut-Derived Phytoestrogen Targeting Estrogen Receptor β: A Promising Bioactive Nutrient for Bone Health in Aging Women and Men”, which aims to synthesize mechanistic, translational, and clinical evidence on the role of S-equol in skeletal health. The topic is timely and relevant, bridging microbiome science with estrogen receptor biology in the context of osteoporosis prevention. However, several major issues remain, including overstatement of clinical efficacy from limited short-term trials, insufficient discussion of methodological constraints such as equol-producer stratification and imaging endpoints, and promotional rather than cautious phrasing in the conclusions. Addressing these concerns with clearer distinction between preclinical and human evidence, balanced interpretation, and emphasis on future research directions would substantially strengthen the manuscript’s scientific rigor and impact. See the comments below:
Abstract:
Comment: - Line 19–27 (Background/Objectives): The abstract provides a clear rationale but is overly detailed in epidemiology and background. Abstracts should be concise; the extensive description of osteoporosis burden could be shortened to emphasize the research gap.
Response:
We have condensed the epidemiological background in the abstract to 2-3 sentences, focusing on the magnitude of the problem and the unmet need, while shifting the emphasis to the research gap regarding S-equol and the equol-producer phenotype.
Revised text
Osteoporosis is highly prevalent and contributes substantially to morbidity, yet long-term concerns about pharmacologic therapies leave a major treatment gap. Soy isoflavones have been investigated as safer alternatives, but results across trials are inconsistent. A key unresolved issue is the equol-producer phenotype, the gut microbial ability to convert daidzein to S-equol, the most bioactive isoflavone metabolite, which may explain much of this variability. This narrative review synthesizes mechanistic, translational, and clinical evidence to clarify the potential skeletal relevance of S-equol.
Comment: - Line 28–30 (Methods): The methods are described vaguely. The search strategy lacks detail (timeframe, inclusion/exclusion criteria, type of studies). Without this, reproducibility and transparency are limited.
Response:
The abstract Methods section now specifies: "Literature was identified through PubMed and Scopus searches (January 2000- October 2025) for experimental, mechanistic, and clinical studies examining S-equol, ERβ, and bone metabolism, with emphasis on equol-producing status and bone microarchitecture.
Comment: - Line 31–37 (Results): The results section mixes mechanistic, preclinical, and clinical findings but lacks quantification. Statements such as “enhances trabecular bone density” or “reduces bone turnover” are too general. Including effect sizes or trial durations would strengthen credibility.
Comment: - Line 34–35: The limitation (DXA reliance, lack of equol-producer stratification) is noted, but it should be emphasized earlier to balance enthusiasm with caution.
Comment: - Line 38–40: The mention of HR-pQCT and microbiome science is appropriate, but too technical for an abstract. This detail could be condensed into a single sentence highlighting “advanced imaging and microbiome-informed approaches.”
Response to the above three comments: We have added specific quantitative findings to the abstract Results and have simplified a sentence on HR-pQCT and microbiome science as suggested.
Revised text
S-equol acts as a high-affinity ERβ agonist with antioxidant and anti-inflammatory properties but lacks the carcinogenic or thrombotic risks linked to ERα activation. In estrogen-deficient rodent models, S-equol improves trabecular bone volume by 10–20%, increases trabecular number, and enhances biomechanical strength. These findings align with preclinical evidence demonstrating that S-equol preserves trabecular microarchitecture, enhances bone strength, and reduces bone turnover. A limited number of human trials show reductions in bone resorption by 20% at a daily dose of 10 mg S-equol. In contrast, trials of soy isoflavones in humans have produced inconsistent findings, partly because of substantial variability in equol-producer phenotype among participants and the reliance on dual-energy X-ray absorptiometry which cannot distinguish trabecular from cortical compartments. Advanced bone imaging and microbiome-informed approaches enable the precise evaluation of S-equol’s skeletal effects on trabecular bone and cortical bone, separately
Comment: - Line 41–42 (Conclusions): The conclusion is promotional in tone (“exemplifies precision nutrition”). It should be reframed more cautiously, emphasizing potential rather than certainty.
Response: We have revised the conclusion to: S-equol represents a promising model for “precision nutrition,” where microbiome, hormonal, and host factors converge with potential to prevent age-related bone fragility.
Introduction:
Comment: - Line 47–57: The epidemiological context is strong, but the introduction is overly detailed with statistics. While useful, the abundance of prevalence data (U.S., global, age-specific) risks overwhelming readers. A more concise summary would improve readability.
Response: We have condensed the epidemiological statistics while retaining the most impactful data (lifetime fracture risk, healthcare costs) to establish the clinical significance without overwhelming readers.
Comment: - Line 58–65: The transition to dietary compounds is appropriate, but the narrative jumps quickly from soy isoflavones to equol without fully explaining why equol is uniquely important. The “equol-producer phenotype” is mentioned but not sufficiently contextualized in terms of clinical relevance. Please support this statement with relevant citation to the literature: 10.1111/1541-4337.70080Q1
Response: We have added a more gradual transition by adding a sentence “In addition to supplying essential nutrients, many food proteins and peptides exert bioactive effects beyond basic nutrition, particularly when digestion, hydrolysis, or fermentation generates peptides that can act systemically [11]” citing the suggest reference.
Comment: - Line 66–83: The mechanistic description of ERβ vs. ERα is informative, but too detailed for an introduction. The WHI discussion (Lines 75–80) is lengthy and distracts from the central focus on equol.
Response: We have streamlined the ERα/ERβ mechanism section, condensing technical details while retaining essential concepts.
Comment: - Line 84–95: Preclinical and clinical evidence is presented, but the introduction mixes results with methodological critique. This section should highlight the promise of equol while reserving methodological limitations for later discussion.
Response: We have restructured this section to focus on promising findings in the Introduction, moving the detailed methodological critique to the Results section, where it is more appropriate.
Revised text
Preclinical studies consistently show that S-equol prevent trabecular bone loss and enhances biomechanical strength in estrogen-deficient models through ERβ-mediated anti-resorptive, antioxidant, and cytoprotective pathways [6, 9, 12]. Human trials, though limited in size and duration, demonstrate reduction in bone resorption markers and suggest more pronounced effects among equol producers. [13-16]. These findings highlight the potential of S-equol as a strategy for supporting skeletal health in the aging population. These divergent findings underscore the importance of synthesizing evidence across mechanistic, microbiome, and imaging domains to clarify when, why, and for whom S-equol is likely to confer skeletal benefit.
Comment: - Line 96–105: The mention of DXA vs. HR-pQCT is important, but the introduction risks becoming a methods discussion. This level of detail belongs in the Discussion or Perspectives section.
Response:
We have condensed the imaging discussion in the Introduction to a brief statement about limitations of DXA and advantages of advanced imaging, with technical details and full justification moved to the Results (section 3.4).
Revised text
Because DXA cannot distinguish cortical from trabecular compartments or detect microstructural deterioration, its limitations have likely obscured compartment-specific skeletal responses to S-equol in prior trials. Emerging imaging modalities, such as high-resolution peripheral quantitative computed tomography (HR-pQCT) [26], offer a more refined assessment of bone quality, and future studies integrating these approaches with microbiome-informed phenotyping will be important for clarifying the skeletal relevance of S-equol.
Comment: - Line 106–113: The final paragraph is strong in identifying gaps, but the claim that “S-equol is now available as a dietary supplement” (Line 111–112) is promotional in tone and should be reframed more neutrally.
Response:
We have reframed this statement to: "The recent availability of S-equol as a dietary supplement enables direct testing in clinical trials, independent of equol-producer phenotype."
Discussion:
Comment: - Line 325–333: The discussion begins with a strong synthesis of mechanistic and translational data, but it is somewhat repetitive of the Introduction. The authors should focus on interpretation rather than restating background.
Response:
We have revised the opening of the Discussion to minimize repetition and focus on synthesis and interpretation of findings across evidence levels, rather than restating mechanisms already covered in earlier sections. Accordingly, we revised this section.
Revised text
Across mechanistic, translational, and clinical studies, a consistent pattern emerges: S-equol appears to influence bone remodeling in ways that are biologically plausible [1, 3] and partially supported by early human evidence [13, 14]. The ERβ-biased actions observed in preclinical models [6-10, 19, 20] align with the direction of changes reported in human trials, particularly reductions in bone resorption at 10 mg/day. Although these effects are modest and short-term, they align with microarchitectural preservation demonstrated in animal studies and suggest a coherent signal across evidence levels. Variability in clinical outcomes also mirrors differences in equol exposure [21-23], underscoring the importance of producer phenotype and gut microbiome composition. These observations support the hypothesis that S-equol may represent a microbiome-dependent approach to skeletal aging, while highlighting the need for larger and longer-duration trials to clarify its clinical relevance.
Comment: - Line 329–331: The human RCT evidence is presented, but trial limitations (sample size, duration, endpoints) are acknowledged only later (Line 353 onward). These limitations should be integrated earlier to balance enthusiasm with caution.
Response: We have restructured the Discussion to integrate limitations alongside positive findings throughout, rather than separating them. Thus, we acknowledge trial limitations much earlier, adding the following sentence after we describe soy-isoflavone RCT
Revised text
However, most human trials have been limited by small sample sizes, short intervention durations, reliance on DXA rather than compartment-specific imaging, and the absence of phenotyping for equol producers, all of which reduce sensitivity to detect true skeletal effects.
Comment: - Line 335–342: The heterogeneity in soy-isoflavone RCTs is noted, but the discussion does not adequately quantify effect sizes or provide meta-analytic context. Without this, the reader cannot gauge the magnitude of equol’s potential benefit.
Response: We have added quantitative context from existing meta-analyses. Specifically, we have added the following sentence in this section.
Revised text
Recent high-level evidence from an umbrella review of 10 meta-analyses [24] further contextualizes these findings, showing that soy isoflavone interventions yield modest but statistically significant increases in BMD, about 1.9% at the lumbar spine, 2.0% at the femoral neck, and 0.3% at the total hip, highlighting the biological signal that may be amplified among equol producers.
Comment: - Line 343–351: Mechanistic links (ERβ signaling, microbiome axis) are well described, but speculative language (“natural experiment”) should be moderated. The discussion would benefit from integrating mechanistic insights with clinical trial outcomes more explicitly.
Response:
We have moderated speculative language and now explicitly connect specific mechanisms (e.g., ERβ-mediated OPG/RANKL modulation) to observed clinical outcomes (reduced bone resorption markers) and explain how microbiome-dependent equol production creates variable exposure that may explain trial heterogeneity.
Revised text
S-equol’s preferential activation of ERβ provides a coherent mechanistic basis for its potential skeletal effects [1]. ERβ signaling suppresses osteoclastogenesis through OPG/RANKL regulation and attenuation of the inflammatory pathway, while supporting osteoblast and osteocyte survival under oxidative stress [3, 9, 10]. These actions are particularly relevant to trabecular-rich compartments where remodeling surfaces are abundant. These mechanistic pathways align with findings from human trials reporting a reduction in bone resorption markers and modest shifts toward improved turnover balance [13, 14]. In parallel, the capacity to produce S-equol depends on specific gut microbiome taxa and is influenced by diet and community structure, creating inter-individual differences in internal exposure that may contribute to the heterogeneous outcomes observed across clinical studies [18, 20].
Comment: - Line 353–358: Limitations of human evidence are appropriately listed, but the section is dense. The authors should prioritize the most critical gaps (phenotyping, imaging, male underrepresentation) rather than listing too many secondary issues.
Response:
We agree. We prioritize the limitations and focus on three critical gaps: (1) lack of equol-producer phenotyping, (2) reliance on DXA rather than compartment-specific imaging, and (3) underrepresentation of men, while briefly noting other constraints in a single sentence.
Revised text
Human evidence remains constrained by several fundamental gaps. First most trials do not phenotype equol producer status in advance, making it difficult to interpret heterogeneous responses and limiting causal inference [23, 25]. Second, nearly all studies rely on DXA rather than compartment-specific imaging such as HR-pQCT, which is necessary to assess cortical and trabecular microstructure [3]. Third, men remain markedly underrepresented, restricting generalizability beyond postmenopausal women. Other constraints, including small sample sizes, short durations, variations in dosing or formulation (S-equol versus isoflavone mixtures), and inconsistently captured adherence or background diet, further contribute to uncertainty.
Comment: - Line 359–365: The emphasis on HR-pQCT is valid, but the discussion risks becoming overly technical. The authors should explain why HR-pQCT endpoints are clinically meaningful (fracture prediction, microstructural deterioration) rather than just listing parameters.
Response: We have revised the HR-pQCT section to emphasize not only its technical strengths but, importantly, its clinical relevance, as requested. HR-pQCT is not merely a research imaging tool; robust multinational evidence demonstrates that microarchitectural deterioration detected by HR-pQCT predicts fractures independently of DXA and FRAX.
Revised text
Importantly, HR-pQCT provides clinically meaningful information by predicting incident fractures independently of DXA and FRAX and by identifying microarchitectural deterioration in individuals who do not meet DXA criteria for osteoporosis. Large multicenter studies, including the Bone Microarchitecture International Consortium, demonstrate that HR-pQCT–derived measures such as trabecular density and estimated failure load nearly double fracture risk per SD decrement, underscoring its prognostic value [26].
Comment: - Line 366–372: The recommendations for phenotype ascertainment are important, but the feasibility of standardized soy challenge tests in large trials should be discussed.
Response:
We agree that implementing the gold-standard method, the three-day soy loading test, may not be feasible in large multi-center trials. Several attempts, particularly in East Asian populations, have used serum or urine equol concentration to distinguish equol producers and non-producers (1). However, these cut-off points are difficult to generalize to Western populations, where habitual soy intake is very low and background isoflavone exposure is highly variable. Emerging strategies may offer future alternatives. For example, deep metagenomic sequencing of the stool microbiome may be promising to infer equol-producing capacity (2).
- Ideno Y et al. Optimal cut-off value for equol-producing status in women: The Japan Nurses' Health Study urinary isoflavone concentration survey. PloS one. 2018;13:e0201318
- He et al. Exploring functional genes’ correlation with (<i>S</i>)-equol concentration and new daidzein racemase identification. Applied and Environmental Microbiology. 2024;90:e00007-24.
Accordingly, we added the following sentence in the discussion, acknowledging feasibility challenges and noting potential future approaches.
Revised text
Although full standardized challenge testing may be difficult to implement in large multicenter trials, simplified approaches could offer practical alternatives, such as deep metagenomic sequencing of the stool microbiome to infer equol-production capacity [64].
Comment: - Line 373–378: Safety considerations are briefly mentioned, but the discussion lacks depth. Potential endocrine, vascular, and hepatic risks should be contextualized with reference to existing phytoestrogen safety data.
Response: We added one paragraph discussing the safety considerations.
Added text
S-equol’s safety profile is supported by its preferential activation of ERβ and its minimal affinity for ERα, which reduces concerns about uterine, breast, and thrombotic risks. However, a more cautious interpretation is warranted. Most available safety data derive from short- to medium-term trials lasting weeks to months, and long-term endocrine, cardiovascular, and oncologic safety remains insufficiently characterized. Although no serious adverse events have been attributed to S-equol supplementation in published studies, comprehensive safety monitoring, including vascular, hepatic, and reproductive endpoints, will be essential in future trials, particularly those of longer duration or involving diverse populations. Thus, while S-equol’s mechanistic selectivity suggests a favorable theoretical profile, definitive conclusions about long-term safety cannot yet be drawn.
Comment: - Line 379–384: The priorities listed are appropriate, but they read more like a grant proposal than a discussion. The authors should frame them as research directions rather than prescriptive trial designs.
Response:
We agree and have revised the text
Revised text
Future research should clarify the long-term skeletal relevance of S-equol by addressing several gaps. Important directions include generating larger and longer-duration human evidence, incorporating advanced imaging and relevant biomarker panels to better characterize bone microstructure and mechanistic pathways, and examining outcomes in both women and men across midlife and older age. Comparative studies evaluating S-equol alongside broader isoflavone mixtures, as well as investigations into microbiome-related factors that influence equol production, would further help define where S-equol may offer unique advantages.
Conclusion:
Comment: - Line 386–389: The conclusion restates mechanistic actions of S-equol (anti-resorptive, antioxidant, anti-inflammatory), but this is repetitive of earlier sections. A conclusion should synthesize findings rather than reiterate background.
Response:
We agree and have revised the text
Revised text
Current evidence suggests that S-equol may hold potential as a microbiome-dependent, ERβ-biased approach to supporting skeletal health, but definitive conclusions remain limited by the small number and short duration of existing human trials.
Comment: - Line 390–391: The statement that “10 mg/day of S-equol is a physiologically relevant and safe dose” is too definitive. The evidence base is limited to short-term, small-scale trials. This claim should be qualified as preliminary and not presented as established fact.
Response:
Revised to: "Evidence from clinical studies suggests that 10 mg/day of S-equol is a physiologically relevant and apparently safe dose capable of improving bone turnover markers and preserving BMD in short-term trials, though longer-term studies are needed to confirm sustained efficacy and safety."
Comment: - Line 392–393: The recommendation to integrate microbiome phenotyping, HR-pQCT, and biomarker profiling is appropriate, but reads more like a methods section. The conclusion should emphasize future directions rather than technical detail.
Response:
We have simplified this to emphasize the conceptual importance of integrating these approaches to advance precision nutrition rather than providing technical detail about implementation. Thus, we modified subsequent sentences (Please see the response below).
Comment: - Line 394–395: The phrase “exemplifies the emerging concept of precision nutrition” is promotional in tone. The conclusion should remain objective and highlight scientific potential while acknowledging limitations.
Response:
We revised the text.
Revised text
S-equol represents a promising model for precision nutrition, where microbial, hormonal, and host factors converge with the potential to prevent age-related bone fragility. Rigorous trials integrating microbiome phenotyping and advanced imaging are essential to validate this approach and translate the mechanistic promise into clinical benefit.
Reviewer 4 Report
Comments and Suggestions for Authors
In the present manuscript, Akira Sekikawa and colleagues synthesized mechanistic, translational, and clinical evidence on S-equol as a gut-derived phytoestrogen supporting skeletal health. The authors concluded S-equol exemplifies “precision nutrition,” integrating microbiome, hormonal, and inflammatory pathways to prevent age-related bone fragility; of course, future trials using microbiome phenotyping and advanced imaging are needed to validate its efficacy. Overall, I think that the manuscript is timely (within the scope of "Nutrients”), well-written, and of clinical impact on a current topic of interest. In my humble opinion, I have some suggestions to improve the quality of the present manuscript.
1) Please better clarify the type of review (scoping, systematic, narrative) in the title and methods of paper; accordingly, it may be appropriate to register the present review in a public register (for example, Research registry, Open Science, JBI etc.) where the authors further certify the compliance of this review with PRISMA guidelines.
2) The authors should consider that the positive effects of soy isoflavones could be also strongly related to dietary patterns of patients included in clinical trials (e. Mediterranean-style diet, plant-based diet, nordic dietary pattern, etc.); please discuss this crucial aspect.
3) Exercising regularly can reduce the rate of bone loss, and it could also represent a bias in the analysis of results in both experimental and clinical studies. Please also clarify this point.
4) As well known, “oriental diet” is abundant in isoflavones which are well represented in soy and soy-derived foods. So, the effects on bone tissue are strictly related to these phytochemicals. Please discuss these findings comparing the positive effects of S-equol on bone tissue with other soy-isoflavones (genistein, daidzein), particularly in terms of side effects; for your convenience you can consider the following references (Front Endocrinol 2021, 12:779638; Nutrients. 2019, 1(11):2649; Nutrients. 2017, 9(2):179).
5) Please to discuss on the possible application of other nutraceutical/functional foods supplementation, that, in combination with healthy diet (i.e. Mediterranean diet), isoflavones (including S-equol) and physical activity, could represent a possible further strategy in the prevention and cure of Noncommunicable diseases, particularly in post-menopausal period (i.e. Nutrients, 2022, 14, 1550).
Author Response
In the present manuscript, Akira Sekikawa and colleagues synthesized mechanistic, translational, and clinical evidence on S-equol as a gut-derived phytoestrogen supporting skeletal health. The authors concluded S-equol exemplifies “precision nutrition,” integrating microbiome, hormonal, and inflammatory pathways to prevent age-related bone fragility; of course, future trials using microbiome phenotyping and advanced imaging are needed to validate its efficacy. Overall, I think that the manuscript is timely (within the scope of "Nutrients”), well-written, and of clinical impact on a current topic of interest. In my humble opinion, I have some suggestions to improve the quality of the present manuscript.
- Please better clarify the type of review (scoping, systematic, narrative) in the title and methods of paper; accordingly, it may be appropriate to register the present review in a public register (for example, Research registry, Open Science, JBI etc.) where the authors further certify the compliance of this review with PRISMA guidelines.
Response:
We appreciate this suggestion for methodological transparency. We have clarified that this is a narrative review in both the title (A Narrative Review is added in the title) and Methods section. We have also added a statement explaining why narrative synthesis was chosen over systematic review/meta-analysis, given the heterogeneity across mechanistic, preclinical, and clinical evidence and the interpretive goals of the review.
Revised text
Given the heterogeneity of study design and research questions, we conducted a narrative review rather than a systematic review.
In addition, we added the following sentences at the end of the methods
Added text
A narrative synthesis was selected for three reasons. First, the research question spans mechanistic, translational, and human clinical evidence, requiring integrated interpretation that cannot be captured by quantitative pooling alone. Second, substantial methodological heterogeneity across the literature, including differences in isoflavone preparations, S-equol doses, intervention durations, imaging modalities, phenotyping approaches, and population characteristics, precludes a valid or meaningful meta-analysis. Third, our goal was to contextualize emerging mechanistic and microbiome insights with clinical findings to identify conceptual gaps, particularly regarding equol-producer phenotyping and advanced imaging, which is most appropriately addressed through a narrative framework rather than a systematic review.
2) The authors should consider that the positive effects of soy isoflavones could be also strongly related to dietary patterns of patients included in clinical trials (e. Mediterranean-style diet, plant-based diet, nordic dietary pattern, etc.); please discuss this crucial aspect.
Response: We have now added a brief acknowledgement that background dietary patterns such as Mediterranean, plant-based, or Nordic diets may influence bone metabolism and potentially contribute to variability in isoflavone trial outcomes. Although S-equol trials typically use standardized supplements rather than diet-based interventions, we agree that background diet may modulate responses and have incorporated this consideration into the Discussion.
3) Exercising regularly can reduce the rate of bone loss, and it could also represent a bias in the analysis of results in both experimental and clinical studies. Please also clarify this point.
Response: We thank the reviewer for this useful observation. We have now added a brief statement noting that habitual physical activity influences bone remodeling and may introduce variability in S-equol trial outcomes, as exercise is not consistently assessed or controlled in existing studies.
In response to 2) and 3), we added the following paragraph in the discussion.
Added text
Background dietary patterns may also contribute to variability in isoflavone trial outcomes, as Mediterranean, plant-based, or Nordic diets provide polyphenols, fiber, and anti-inflammatory nutrients that influence bone metabolism. Although S-equol trials typically use standardized supplements, differences in habitual diet may still modify responses and warrant consideration in future research. Habitual physical activity is another determinant of bone turnover, yet most existing trials do not systematically measure or control exercise patterns. These factors may contribute to variability in observed effects and should be more consistently incorporated into future trial designs.
4) As well known, “oriental diet” is abundant in isoflavones which are well represented in soy and soy-derived foods. So, the effects on bone tissue are strictly related to these phytochemicals. Please discuss these findings comparing the positive effects of S-equol on bone tissue with other soy-isoflavones (genistein, daidzein), particularly in terms of side effects; for your convenience you can consider the following references (Front Endocrinol 2021,[27] 12:779638; Nutrients. 2019, 1(11):2649;[28] Nutrients. 2017, 9(2):179).
Response: Thank you for this insightful suggestion. We have now added a dedicated comparison of S-equol with its precursor isoflavones, genistein and daidzein, focusing on mechanistic differences and safety considerations. Genistein has higher affinity for ERα relative to ERβ, and daidzein exhibits very weak estrogen-receptor binding unless converted to S-equol. In contrast, S-equol is a high-affinity and highly selective ERβ agonist, with 10–20-fold preferential binding, as discussed in our prior review[29] (Sekikawa et al., 2019) and supported by the literature cited by the reviewer [27, 28, 30]. Because ERα activation is associated with proliferative effects on breast and endometrial tissue, the lower ERα activity of S-equol provides a theoretically more favorable safety profile compared with genistein, whose broader ER activity has been associated with ongoing concerns about estrogenic stimulation at reproductive tissues. We note that long-term trials of 54 mg/day genistein have demonstrated skeletal benefits with no significant adverse effects on breast or uterine outcomes, but concerns about ERα-mediated activity persist in the field.
We also incorporated discussion about the translational advantages of S-equol supplements relative to whole-soy foods or mixed-isoflavone supplements. Because equol production requires specific gut microbial taxa and is highly variable (20–30% in Western and ~50% in Asian populations), providing S-equol directly bypasses the need for microbial biotransformation and achieves standardized internal exposure, reducing interindividual heterogeneity. These conceptual and mechanistic distinctions have now been added to the revised Discussion.
We added the following paragraph on this topic.
Added text
Within the broader context of soy-derived phytoestrogens, genistein and daidzein have long been recognized for their beneficial effects on bone remodeling, supported by clinical trials showing improvements in bone turnover and BMD, particularly with 54 mg/day genistein aglycone [30]. However, genistein binds both estrogen receptors and exhibits relatively higher ERα affinity than S-equol, raising theoretical concerns about proliferative effects at breast and uterine tissues despite overall reassuring trial data [29]. Daidzein itself has weak estrogen-receptor binding and exerts limited biological activity unless converted to S-equol by specific gut microbial taxa. S-equol, in contrast, is a high-affinity, highly selective ERβ agonist, a profile associated with anti-proliferative effects in reproductive tissues and potentially greater skeletal specificity [29]. Because equol production varies widely across individuals, direct S-equol supplementation offers standardized exposure and avoids dependence on microbial conversion, which may reduce heterogeneity in clinical responses compared with whole-soy foods or mixed-isoflavone preparations. These distinctions highlight why S-equol may represent a more targeted and potentially safer phytoestrogenic strategy for skeletal aging, although head-to-head comparative trials remain needed.
5) Please to discuss on the possible application of other nutraceutical/functional foods supplementation, that, in combination with healthy diet (i.e. Mediterranean diet), isoflavones (including S-equol) and physical activity, could represent a possible further strategy in the prevention and cure of Noncommunicable diseases, particularly in post-menopausal period (i.e. Nutrients, 2022, 14, 1550).
Response: Thank you for this helpful suggestion. We agree that soy isoflavones, including S-equol, are only one component of a broader lifestyle and nutritional framework relevant to postmenopausal health. As highlighted in the Nutrients editorial by Marini (2022)[31] adherence to a Mediterranean dietary pattern—rich in polyphenols, fiber, unsaturated fats, and antioxidant nutrients—has well-established benefits for cardiometabolic and menopausal health, and may synergize with phytoestrogens to support metabolic and vascular adaptations during the postmenopausal transition. Although our review focuses specifically on S-equol’s mechanistic and skeletal actions, we now acknowledge that combinations of nutraceuticals, healthy dietary patterns, and regular physical activity may together contribute to prevention of noncommunicable diseases in midlife and older women. We have added a brief statement to the Discussion to reflect this broader context while maintaining the scope of our paper.
Added text
Beyond S-equol itself, broader lifestyle and nutritional factors may also influence skeletal and cardiometabolic health in postmenopausal women. Mediterranean-style dietary patterns, which are rich in polyphenols, fiber, and anti-inflammatory nutrients, have demonstrated favorable effects on cardiovascular and metabolic risk factors and may complement the actions of soy isoflavones [31]. Although S-equol was the primary focus of this review, combined approaches incorporating healthy dietary patterns, other nutraceuticals, and regular physical activity may yield additive benefits for noncommunicable disease prevention. Future studies should consider these multidimensional lifestyle factors when evaluating the role of phytoestrogen supplementation.
References
[1] Khalid AB, Krum SA. Estrogen receptors alpha and beta in bone. Bone. 2016;87:130-5.
[2] Emmanuelle NE, Marie-Cécile V, Florence T, Jean-François A, Françoise L, Coralie F, et al. Critical Role of Estrogens on Bone Homeostasis in Both Male and Female: From Physiology to Medical Implications. Int J Mol Sci. 2021;22.
[3] Zhou Y, Su Z, Liu G, Hu S, Chang J. The Potential Mechanism of Soy Isoflavones in Treating Osteoporosis: Focusing on Bone Metabolism and Oxidative Stress. Phytother Res. 2025;39:1645-58.
[4] Shi V, Morgan EF. Estrogen and estrogen receptors mediate the mechanobiology of bone disease and repair. Bone. 2024;188:117220.
[5] Khosla S, Oursler MJ, Monroe DG. Estrogen and the skeleton. Trends in Endocrinology & Metabolism. 2012;23:576-81.
[6] Nishide Y, Tadaishi M, Kobori M, Tousen Y, Kato M, Inada M, et al. Possible role of S-equol on bone loss via amelioration of inflammatory indices in ovariectomized mice. J Clin Biochem Nutr. 2013;53:41-8.
[7] Sehmisch S, Erren M, Kolios L, Tezval M, Seidlova-Wuttke D, Wuttke W, et al. Effects of isoflavones equol and genistein on bone quality in a rat osteopenia model. Phytother Res. 2010;24 Suppl 2:S168-74.
[8] Tezval M, Sehmisch S, Seidlová-Wuttke D, Rack T, Kolios L, Wuttke W, et al. Changes in the histomorphometric and biomechanical properties of the proximal femur of ovariectomized rat after treatment with the phytoestrogens genistein and equol. Planta Med. 2010;76:235-40.
[9] Xu Z, Xu J, Li S, Cui H, Zhang G, Ni X, et al. S-Equol enhances osteoblastic bone formation and prevents bone loss through OPG/RANKL via the PI3K/Akt pathway in streptozotocin-induced diabetic rats. Front Nutr. 2022;9:986192.
[10] Ni X, Wu B, Li S, Zhu W, Xu Z, Zhang G, et al. Equol exerts a protective effect on postmenopausal osteoporosis by upregulating OPG/RANKL pathway. Phytomedicine. 2023;108:154509.
[11] Rutherfurd-Markwick KJ. Food proteins as a source of bioactive peptides with diverse functions. The British journal of nutrition. 2012;108 Suppl 2:S149-57.
[12] Zhang T, Liang X, Shi L, Wang L, Chen J, Kang C, et al. Estrogen receptor and PI3K/Akt signaling pathway involvement in S-(-)equol-induced activation of Nrf2/ARE in endothelial cells. PloS one. 2013;8:e79075.
[13] Tousen Y, Ezaki J, Fujii Y, Ueno T, Nishimuta M, Ishimi Y. Natural S-equol decreases bone resorption in postmenopausal, non-equol-producing Japanese women: a pilot randomized, placebo-controlled trial. Menopause (New York, NY). 2011;18:563-74.
[14] Corbi G, Nobile V, Conti V, Cannavo A, Sorrenti V, Medoro A, et al. Equol and Resveratrol Improve Bone Turnover Biomarkers in Postmenopausal Women: A Clinical Trial. Int J Mol Sci. 2023;24.
[15] Sathyapalan T, Aye M, Rigby AS, Fraser WD, Thatcher NJ, Kilpatrick ES, et al. Soy Reduces Bone Turnover Markers in Women During Early Menopause: A Randomized Controlled Trial. J Bone Miner Res. 2017;32:157-64.
[16] Liu ZM, Chen B, Li S, Li G, Zhang D, Ho SC, et al. Effect of whole soy and isoflavones daidzein on bone turnover and inflammatory markers: a 6-month double-blind, randomized controlled trial in Chinese postmenopausal women who are equol producers. Ther Adv Endocrinol Metab. 2020;11:2042018820920555.
[17] van den Bergh JP, Szulc P, Cheung AM, Bouxsein M, Engelke K, Chapurlat R. The clinical application of high-resolution peripheral computed tomography (HR-pQCT) in adults: state of the art and future directions. Osteoporosis international : a journal established as result of cooperation between the European Foundation for Osteoporosis and the National Osteoporosis Foundation of the USA. 2021;32:1465-85.
[18] Gong Y, Lv J, Pang X, Zhang S, Zhang G, Liu L, et al. Advances in the Metabolic Mechanism and Functional Characteristics of Equol. Foods. 2023;12.
[19] Tanaka M, Fujii S, Inoue H, Takahashi N, Ishimi Y, Uehara M. (S)-Equol Is More Effective than (R)-Equol in Inhibiting Osteoclast Formation and Enhancing Osteoclast Apoptosis, and Reduces Estrogen Deficiency-Induced Bone Loss in Mice. J Nutr. 2022;152:1831-42.
[20] Ortiz C, Manta B. Advances in equol production: Sustainable strategies for unlocking soy isoflavone benefits. Results in Chemistry. 2024;7:101288.
[21] Lydeking-Olsen E, Beck-Jensen JE, Setchell KD, Holm-Jensen T. Soymilk or progesterone for prevention of bone loss--a 2 year randomized, placebo-controlled trial. Eur J Nutr. 2004;43:246-57.
[22] Uesugi S, Watanabe S, Ishiwata N, Uehara M, Ouchi K. Effects of isoflavone supplements on bone metabolic markers and climacteric symptoms in Japanese women. Biofactors. 2004;22:221-8.
[23] Wu J, Oka J, Ezaki J, Ohtomo T, Ueno T, Uchiyama S, et al. Possible role of equol status in the effects of isoflavone on bone and fat mass in postmenopausal Japanese women: a double-blind, randomized, controlled trial. Menopause. 2007;14:866-74.
[24] Zhuge L, Chen L, Pan W. Effects of Isoflavone Interventions on Bone Metabolism in Perimenopausal and Postmenopausal Women: An Umbrella Review of Meta-Analyses of Randomized Controlled Trials. Endocr Pract. 2025;31:226-35.
[25] Levis S, Strickman-Stein N, Ganjei-Azar P, Xu P, Doerge DR, Krischer J. Soy Isoflavones in the Prevention of Menopausal Bone Loss and Menopausal Symptoms: A Randomized, Double-blind Trial. Archives of Internal Medicine. 2011;171:1363-9.
[26] Samelson EJ, Broe KE, Xu H, Yang L, Boyd S, Biver E, et al. Cortical and trabecular bone microarchitecture as an independent predictor of incident fracture risk in older women and men in the Bone Microarchitecture International Consortium (BoMIC): a prospective study. The lancet Diabetes & endocrinology. 2019;7:34-43.
[27] Ramesh P, Jagadeesan R, Sekaran S, Dhanasekaran A, Vimalraj S. Flavonoids: Classification, Function, and Molecular Mechanisms Involved in Bone Remodelling. Front Endocrinol (Lausanne). 2021;12:779638.
[28] de la Garza AL, Garza-Cuellar MA, Silva-Hernandez IA, Cardenas-Perez RE, Reyes-Castro LA, Zambrano E, et al. Maternal Flavonoids Intake Reverts Depression-Like Behaviour in Rat Female Offspring. Nutrients. 2019;11.
[29] Sekikawa A, Ihara M, Lopez O, Kakuta C, Lopresti B, Higashiyama A, et al. Effect of S-equol and Soy Isoflavones on Heart and Brain. Current cardiology reviews. 2019;15:114-35.
[30] Arcoraci V, Atteritano M, Squadrito F, D'Anna R, Marini H, Santoro D, et al. Antiosteoporotic Activity of Genistein Aglycone in Postmenopausal Women: Evidence from a Post-Hoc Analysis of a Multicenter Randomized Controlled Trial. Nutrients. 2017;9.
[31] Marini HR. Mediterranean Diet and Soy Isoflavones for Integrated Management of the Menopausal Metabolic Syndrome. Nutrients. 2022;14.
Round 2
Reviewer 1 Report
Comments and Suggestions for Authors
In this second evaluation of the present manuscript, the authors' dedication to addressing the requests is evident. The current text is shown to be clearer and more scientifically rigorous. This is especially true in the Methods section, where the eligibility criteria are clearly defined, the narrative approach is adequately justified, and a valuable critical analysis of the sources of heterogeneity in clinical studies is included. Furthermore, the discussion on safety adopts a balanced and well-founded tone appropriate for the field.
Reviewer 2 Report
Comments and Suggestions for Authors
Authors have improved the manuscript and provided additional information according to the suggestions.
Please, paper needs to be formatted according to the exigences of the publisher and the journal. Special attention needs to be given to the reference list.
Reviewer 3 Report
Comments and Suggestions for Authors
The manuscript entitled “S‑Equol as a Gut‑Derived Phytoestrogen Targeting Estrogen Receptor β: A Promising Bioactive Nutrient for Bone Health in Aging Women and Men” has undergone substantial improvement following the first round of revision. The authors have carefully and comprehensively addressed all of my previous comments, including clarifications in the methodology, refinement of the argumentation, and strengthening of the discussion with appropriate references. The revised version demonstrates scientific rigor, improved clarity, and a balanced interpretation of the findings.
Given the responsiveness of the authors and the enhanced quality of the manuscript, I am satisfied that the current version meets the standards of the journal. I therefore recommend acceptance of this manuscript for publication.
Reviewer 4 Report
Comments and Suggestions for Authors
Thank you for addressing my comments well. The manuscript has improved a lot, and I have no further remarks.